# Global Simulation of Semivolatile Organic Compounds – Development and Evaluation of the MESSy Submodel SVOC (v1.0)

**Mega Octaviani**[1]
**Holger Tost**[2]
**Gerhard Lammel**[1,3,*]

[1]*Multiphase Chemistry Department, Max Planck Institute for Chemistry, 55128 Mainz, Germany*
[2]*Institute for Atmospheric Physics, Johannes Gutenberg-University Mainz, 55099 Mainz, Germany*
[3]*Research Centre for Toxic Compounds in the Environment, Masaryk University, 62500 Brno, Czech Republic*

[*]Correspondence to: g.lammel@mpic.de

Submitted to Geoscientific Model Development
Manuscript type: Model description

## Abstract

The new submodel SVOC for the Modular Earth Submodel System (MESSy) was developed and applied within the ECHAM5/MESSy Atmospheric Chemistry (EMAC) model to simulate the atmospheric cycling and air-surface exchange processes of semivolatile organic pollutants. Our focus is on four polycyclic aromatic hydrocarbons (PAHs) of largely varying properties. Some new features in input and physics parameterizations of tracers were tested: emission seasonality, the size discretization of particulate-phase tracers, the application of poly-parameter linear free energy relationships in gas-particle partitioning, and re-volatilization from land and sea surfaces. The results indicate that the predicted global distribution of the 3-ring PAH phenanthrene is sensitive to the seasonality of its emissions, followed by the effects from considering re-volatilization from surfaces. The predicted distributions of the 4-ring PAHs fluoranthene and pyrene, and the 5-ring PAH benzo(a)pyrene are found sensitive to the combinations of factors with their synergistic effects being stronger than the direct effects of the individual factors. The model was validated against observations of PAH concentrations and aerosol particulate mass fraction. The annual mean concentrations are simulated to the right order of magnitude for most cases and the model well captures the species and regional variations. However, large underestimation is found over the ocean. It is found that the particulate mass fraction of the benzo(a)pyrene is well simulated whereas those of other species are lower than observed.

## 1   Introduction

The atmospheric cycling of semivolatile organic compounds (SOCs) is particularly complex, because of partitioning across phases and air−surface exchange processes, including multihopping (or 'grasshopper effect'; Semeena and Lammel, 2005) and accumulation in ground compartments such as seawater, soil, vegetation, and ice/snow. Many SOCs do resist degradation in environmental compartments, hence, are persistent. In regulation of chemical substances and in international chemicals legislation (e.g. UNEP, 2017), model-based quantifications of the overall environmental residence time (persistence) and the long-range transport potential are requested or encouraged to be applied.

Global and regional distribution and transport of SOCs has been studied using multimedia fate (box) models and chemistry transport models (CTMs) (Scheringer and Wania, 2003). The multimedia models describe the whole or part of the globe as a few zones of homogeneous environmental characteristics (Wania and Mackay, 1999; Mackay, 2010). These models are used as tools to assess the influences of environmental parameters and change on pollutant levels in multiple compartments (Dalla Valle et al., 2007; MacLeod et al., 2005; Lamon et al., 2009). On the other hand, CTMs generally imply the application of three-dimensional Eulerian models coupled with surface and chemistry modules (e.g. Ma et al., 2003; Hansen et al., 2004; Malanichev et al., 2004; Gusev et al., 2005; Semeena et al., 2006; Gong et al., 2007;

Friedman and Selin, 2012; Galarneau et al., 2014; Shrivastava et al., 2017). The addition of a surface module aims to describe air−surface exchange processes and biogeochemical cycles of contaminants whereas a chemistry module describes the changes in air concentrations due to phase partitioning and chemical transformations. Compared to the multimedia models, CTMs have better spatial and temporal resolution but require more computational effort. They are suitable for use to investigate the variability and episodic character of environmental fate and transport. To date, pollutants addressed in model studies were persistent organic pollutants, such as dichlorodiphenyltrichloroethane (DDT), polychlorinated biphenyls (PCBs), hexachlorocyclohexanes (HCHs), polycyclic aromatic hydrocarbons (PAHs), and more recent so-called emerging pollutants (e.g. MacLeod et al., 2011).

The sensitivity of distributions to specific processes of SOC cycling and related input parameters has been the focus of CTM-based studies (Semeena et al., 2006; Sehili and Lammel, 2007; Friedman and Selin, 2012; Galarneau et al., 2014; Thackray et al., 2015). Sehili and Lammel (2007), for instance, suggest that the gas−particle partitioning and particulate-phase oxidation scenarios have significant influences on the long-range atmospheric transport of PAHs. This finding is supported by Friedman and Selin (2012), who, furthermore, concluded that the effects are higher than those of irreversible partitioning and of increased aerosol concentrations.

This study presents the new multicompartment module (submodel) SVOC for the Modular Earth Submodel System (MESSy; Jöckel et al., 2006, 2010). MESSy provides a modular framework for simulations accounting for various degree of complexity and to facilitate a continuous future submodel improvements. The submodel has been applied using the general circulation model ECHAM5 (Roeckner et al., 2003, 2006) as a base model. In connection with the ECHAM5/MESSy (EMAC) model, SVOC encompasses a 3D atmosphere and 2D surface compartments (soil, vegetation, snow, and ocean mixed layer), and considers multicompartment fate and exchange processes, such as emission, phase partitioning, wet and dry deposition of gases and particles, degradation, and air−surface gas exchange, including re-volatilization. SVOC is developed and intended to be applied for the study of all potentially re-volatilizing and gas-particle partitioning (hence, semivolatile) compounds. Nevertheless, the focus of this submodel development is the global distribution of four PAH species of largely varying properties. PAHs enter the atmospheric environment as by-products of all technological combustion processes (Shen et al., 2013) and of open fires (Gullett et al., 2008). They are ubiquitous pollutants of particular environmental and health concern (WHO, 2003; Laender et al., 2011; Lammel, 2015) and due to their continuous global emissions. Here we describe submodel development, compare the results to observations, and assess the significance of four model features to PAH distributions and fate. These features are the temporal resolution of emissions, the size discretization of particulate-phase tracers (bulk or modal), the choice of the gas−particle partitioning scheme, and re-volatilization from surfaces.

## 2 Methods

### 2.1 Model descriptions

The global model applied in this study is the ECHAM/MESSy Atmospheric Chemistry−Climate model (EMAC), a three-dimensional Eulerian model for the simulations of meteorological variables, gaseous, aerosols, clouds, and other climate-related parameters. EMAC combines the general circulation model ECHAM5 (here, version 5.3.02) (Roeckner et al., 2003, 2006) with the Modular Earth Submodel System (MESSy version 2.50; Jöckel et al., 2006, 2010). The atmospheric component ECHAM5 derives four prognostic variables, namely, vorticity, divergence, temperature, and the logarithm of surface pressure in truncated series of spherical harmonics, whereas specific humidity, cloud water, and cloud ice are represented in grid point space. MESSy provides a modular framework to define atmospheric dynamics, chemistry, transport, and radiative transfer processes. For a more detailed description of the EMAC model, evaluation and relevant studies, refer to Jöckel et al. (2006, 2010) and http://www.messy-interface.org. List of MESSy process-based modules, hereinafter submodels, applied in the study are summarized in Table 1.

The new MESSy submodel SVOC for simulating the fate and cycling of SOCs in the global environment is presented. Processes involved in the submodel include gas−particle partitioning, volatilization from the surface, dry and wet depositions, chemical and biotic degradations. These processes are connected to other MESSy submodels. For example, deposition of gas-phase SOCs are calculated by the submodels SCAV and DDEP, aerosol microphysics by GMXe, gas-phase chemistry mechanisms by MECCA, and ocean−air flux exchange by AIRSEA. Figure 1 illustrates the SVOC structure within EMAC system and its interactions with other MESSy submodels. More details on some process parameterizations are given in the following section. A user manual can be found in the Supplement with the list of submodel input and output variables.

Insert Table 1
Insert Figure 1

### 2.2 Parameterizations of cycling processes in multiple compartments

#### 2.2.1 Representation of SOC in particulate phase

The parameterizations of aerosol microphysical processes for SOCs such as gas-to-particle partitioning and dry and wet deposition depend on the way the particulate phase is represented in the model. Here, there are two approaches employed in the submodel to represent the particulate-phase SOC: 1) it is assumed as a bulk species, or 2) the particle sizes are resolved into $n$ continuous (modal) distributions. The former will be hereinafter referred to as the *bulk* scheme, and the latter referred to as the *modal* scheme. In the *modal* scheme, $n$ is equal to the 7 log-normal modes of the GMXe submodel, four with hydrophilic coating (*ns*:

nucleation soluble, *ks*: Aitken soluble, *as*: accumulation soluble, *cs*: coarse soluble) and three hydrophobic (*ki*: Aitken insoluble, *ai*: accumulation insoluble, *ci*: coarse insoluble) (Pringle et al., 2010). Each mode is treated as an individual tracer.

### 2.2.2 Partitioning between gas phase and particulate phase

Gas−particle partitioning is assumed to take place when SOC is at equilibrium between the gas and particulate phases. The concentration of the species that is bound to particles ($C_{\text{particle}}$) is calculated with

$$C_{\text{particle}} = \theta \times (C_{\text{particle}} + C_{\text{gas}}) \,, \tag{1}$$

and the particulate mass fraction ($\theta$) is defined as

$$\theta = \frac{C_{\text{particle}}}{C_{\text{particle}} + C_{\text{gas}}} = \frac{K_{\text{p}} \times C_{\text{PM}}}{1 + K_{\text{p}} \times C_{\text{PM}}} = \frac{K_{\text{p}}'}{1 + K_{\text{p}}'} \,, \tag{2}$$

where $C_{\text{PM}}$ is the concentration of particulate matter or PM ($\mu$g m$^{-3}$), $K_{\text{p}}$ is the temperature-dependent particle−air partition coefficient (m$^3$ $\mu$g$^{-1}$), and $K_{\text{p}}'$ is the dimensionless $K_{\text{p}}$.

In a model configuration using size-resolved particles (viz. the *modal* scheme), each SOC tracer is introduced in the model as eight different species, seven aerosol particles in *ns*, *ks*, *as*, *cs*, *ki*, *ai*, *ci* modes and one in the gas phase. The particulate fraction of the species in mode $i$ ($\theta_i$) is calculated using Equation 2 with $C_{\text{PM}_i}$ and $K_{\text{p}_i}$ being the PM mass concentration and aerosol−air partition coefficient in the corresponding mode, respectively. The gaseous concentration $C_{\text{gas}}$ is calculated using the sum of $K_{\text{p}}$ values across modes, as well as total $C_{\text{particle}}$ and $C_{\text{PM}}$:

$$C_{\text{gas}} = \frac{\sum\limits_{i=1}^{7} C_{\text{particle}_i} \Big/ \sum\limits_{i=1}^{7} C_{\text{PM}_i}}{\sum\limits_{i=1}^{7} K_{\text{p}_i}} \,. \tag{3}$$

It is noted that this approach may not hold the constraint of mass consistency, and is thus subject to further corrections. For the current study, the effects from this problem are expected to be minimal, given the fact that PAHs in the particulate phase are mainly distributed in the accumulation mode (Lammel et al., 2010, and references therein).

For $K_{\text{p}}$ calculation four options of gas−particle partitioning schemes are available in SVOC, they are: (1) a parameterization that is based on adsorption onto aerosol surface (Junge, 1977; Pankow, 1987), (2) absorption into organic matter (Finizio et al., 1997), (3) a combination of two ways of organic matter absorption and black carbon adsorption (Lohmann and Lammel, 2004), and (4) multiple phase of the two-ways sorption system (Goss and Schwarzenbach, 2001; Endo and Goss, 2014; Shahpoury et al., 2016). Two schemes used in this study are described below.

**Lohmann−Lammel scheme**

The Lohmann−Lammel scheme takes into account an adsorption onto black carbon (BC) surface in addition to absorption into OM (Lohmann and Lammel, 2004). This dual sorption theory empirically calculates $K_p$ according to the following relation

$$K_{\mathrm{p}} = 10^{-12} \left( f_{\mathrm{OM}} \; \frac{\gamma_{\mathrm{oct}} \; \mathrm{MW}_{\mathrm{oct}}}{\gamma_{\mathrm{OM}} \; \mathrm{MW}_{\mathrm{OM}} \; \rho_{\mathrm{oct}}} \; K_{\mathrm{oa}} + f_{\mathrm{BC}} \; \frac{a_{\mathrm{atm-BC}}}{a_{\mathrm{soot}} \; \rho_{\mathrm{BC}}} \; K_{\mathrm{sa}} \right) \;, \tag{4}$$

where $\rho_{\mathrm{BC}}$ is the density of BC (assumed as 1 kg L$^{-1}$), $\rho_{\mathrm{oct}}$ is the density of octanol (0.82 kg L$^{-1}$ at 20°), $K_{\mathrm{sa}}$ is the partition coefficient between diesel soot and air, $a_{\mathrm{atm-BC}}$ is the available surface of atmospheric BC (m$^2$ g$^{-1}$), and $a_{\mathrm{soot}}$ is the specific surface area of diesel soot (m$^2$ g$^{-1}$). The adsorptive properties of diesel soot are selected to represent the atmospheric BC because this material is considered the most significant type of BC in polluted air.

The $K_{\mathrm{sa}}$ value is calculated as a function of sub-cooled liquid vapor pressure $p_{\mathrm{L}}^{0}$ using an estimate suggested by van Noort (2003),

$$\log K_{\mathrm{sa}} = -0.85 \log p_{\mathrm{L}}^{0} + 8.94 - \log \left( \frac{998}{a_{\mathrm{soot}}} \right) \;, \tag{5}$$

where $a_{\mathrm{soot}}$ in the model is set as 18.21 m$^2$ g$^{-1}$.

**Poly-Parameter Linear Free Energy Relationships (ppLFER) scheme**

The concept of poly-parameter linear free energy relationships (ppLFER) for the prediction

of equilibrium partition coefficients is introduced by Goss and Schwarzenbach (2001), and its application in environmental chemistry has been reviewed by Endo and Goss (2014). This approach can describe a composite of different types of interactions between gas-phase species and aerosols. In contrast, single-parameter LFERs only correlates the partition coefficient to the sub-cooled liquid vapor pressure or the octanol−air partition coefficient of the species,

hence only valid within the group of compounds for which they were developed.

In the study, ppLFER scheme is incorporated into SVOC in which it defines $K_p$ as the sum of individual partition coefficients representing surface adsorption and bulk-phase absorption processes to inorganic and organic aerosols. The formulation of $K_p$ is adopted from Shahpoury et al. (2016) and is described as follows

$$K_{\mathrm{p}} = \frac{K_{\mathrm{p}}'}{C_{\mathrm{PM}}}$$

$$
\begin{aligned}
K_{\mathrm{p}}' = {} & K_{\mathrm{EC}} \times a_{\mathrm{EC}} \times C_{\mathrm{EC}} \times 10^{-6} + \\
& K_{(\mathrm{NH_4})_2\mathrm{SO_4}} \times a_{(\mathrm{NH_4})_2\mathrm{SO_4}} \times C_{(\mathrm{NH_4})_2\mathrm{SO_4}} \times 10^{-6} + \\
& K_{\mathrm{NaCl}} \times a_{\mathrm{NaCl}} \times C_{\mathrm{NaCl}} \times 10^{-6} + \\
& K_{\mathrm{DMSO}} \times \frac{C_{\mathrm{WSOM}}}{\rho_{\mathrm{DMSO}}} \times 10^{-6} + \\
& K_{\mathrm{PU}} \times 0.2 \times C_{\mathrm{WIOM}} \times 10^{-12} + \\
& K_{\mathrm{hexadecane}} \times 0.8 \times \frac{C_{\mathrm{WIOM}}}{\rho_{\mathrm{hexadecane}}} \times 10^{-6}
\end{aligned}
\tag{6}
$$

where $K_{\mathrm{EC}}$, $K_{(\mathrm{NH_4})_2\mathrm{SO_4}}$, and $K_{\mathrm{NaCl}}$ are the substance partition (adsorption) coefficients ($\mathrm{m_{air}^3 \; m_{surface}^{-2}}$) for elemental−carbon/diesel soot, ammonium sulfate, and sodium chloride aerosol surface−air systems, respectively. $K_{\mathrm{DMSO}}$ is the substance partition (absorption) coefficient for dimethyl sulfoxide−air system ($\mathrm{L_{air} \; L_{DMSO}^{-1}}$). $K_{\mathrm{PU}}$ is the substance partition (absorption) coefficient for polyurethane−air system ($\mathrm{m_{air}^3 \; kg_{PU}^{-1}}$). $K_{\mathrm{hexadecane}}$ is the substance partition (absorption) coefficient for hexadecane−air system ($\mathrm{L_{air} \; L_{hexadecane}^{-1}}$). $a_{\mathrm{EC}}$, $a_{(\mathrm{NH_4})_2\mathrm{SO_4}}$, and $a_{\mathrm{NaCl}}$ are the adsorbent specific surface areas: 18.21, 0.1, and 0.1 $\mathrm{m_{surface}^2 \; g_{adsorbent}^{-1}}$, respectively. $\rho_{\mathrm{DMSO}}$ and $\rho_{\mathrm{hexadecane}}$ are the dimethly sulfoxide and hexadecane densities: $1.1\times10^6$ and $0.77\times10^6$ g m$^{-3}$, respectively. $C_{\mathrm{EC}}$, $C_{(\mathrm{NH_4})_2\mathrm{SO_4}}$, $C_{\mathrm{NaCl}}$, $C_{\mathrm{WSOM}}$, $C_{\mathrm{WIOM}}$ are the concentration ($\mathrm{\mu g_{substance} \; m_{air}^{-3}}$) of elemental carbon (here, black carbon), ammonium sulfate, sodium chloride, water-soluble organic matter, and water-insoluble organic matter, respectively.

The ppLFER scheme calculates the sorptive partition coefficient for every aerosol system, as summarized in Table S1. Each coefficient requires information on system parameters ($e$, $s$, $a$, $b$, $v$, $l$), and the constant $c$, as shown in Table S2. The Abraham solute descriptors ($E$, $S$, $A$, $B$, $V$, and $L$) are substance specific, and for the species selected in this study, refer to Table S6. All the predicted partition constants are adjusted to environmental temperature using the van't Hoff equation

$$\ln K_{(\mathrm{T})} = \ln K_{(\mathrm{T_0})} - \frac{\Delta \mathrm{H}}{R}\left(\frac{1}{\mathrm{T}} - \frac{1}{\mathrm{T_0}}\right) \; , \tag{7}$$

where $\Delta \mathrm{H}$ is the enthalpy of solvent-air phase transfer in J mol$^{-1}$. This variable is system specific and calculated by applying the ppLFER equations given in Table S3 and input parameters given in Table S4. The sequence of $K_{\mathrm{p}}$ calculation from ppLFER analysis in SVOC is illustrated in Figure S1.

### 2.2.3 Volatilization

#### Soil

For soil volatilization, two parameterization schemes are implemented in the SVOC submodel, that is, the Jury scheme (Jury et al., 1983, 1990), and the Smit scheme (Smit et al., 1997). The latter was applied in the study which is based on the volatilization of pesticides from the

surface of fallow soils (Smit et al., 1997). The volatilization occurs upon partitioning over three soil phases (solid, gas, and liquid). The concentration of the chemical in the soil system (kg m$^{-3}$) is formulated as

$$C_{\text{soil}} = Q \times C_{\text{vapor}} \; , \tag{8}$$

and the capacity factor $Q$ is given by

$$Q = \psi + \varphi K_{\text{wa}} + \rho_{\text{soil}} K_{\text{wa}} K_{\text{sl}} \; , \tag{9}$$

where $\psi$ and $\varphi$ are the volume fractions of air and moisture, respectively, $\rho$soil is the soil density, $K_{\text{wa}}$ is the water$-$air partition coefficient where $K_{\text{wa}} = 1/K_{\text{aw}}$, and $K_{\text{sl}}$ is the solid$-$liquid partition coefficient. $K_{\text{aw}}$ is calculated based on the Henry's Law constant:

$$K_{\text{aw}} = 1/\left(H \; R \; T\right) \; , \; \text{and} \tag{10}$$

$$H = k_{\text{H}}^{\ominus} \times \exp\left[\frac{-\Delta H_{\text{soln}}}{R}\left(\frac{1}{T} - \frac{1}{T_0}\right)\right] \; , \tag{11}$$

where $H$ is the temperature-adjusted Henry coefficient (M atm$^{-1}$), $R$ is the dry air gas constant ($= 8.314$ J mol K$^{-1}$), $T$ is the environment temperature (K), $\Delta H_{\text{soln}}$ is the enthalpy of dissolution (J mol$^{-1}$).

$K_{\text{sl}}$ can be set equal to the sorption coefficient to soil organic matter $K_{\text{om}}$ times the fraction of organic carbon in soil $f_{\text{OM}_{\text{s}}}$. Since $K_{\text{om}}$ data is not available, the coefficient for sorption to soil organic carbon $K_{\text{oc}}$ was used to estimate $K_{\text{sl}}$:

$$K_{\text{sl}} = 0.56 K_{\text{oc}} f_{\text{OM}_{\text{s}}} \tag{12}$$

Mackay and Boethling (2000) has suggested a reasonably good regression relationship between $K_{\text{oc}}$ and octanol$-$water partition coefficient $K_{\text{ow}}$ for PAHs:

$$\log(K_{\text{oc}}/1000) = 0.823 \log(K_{\text{ow}}/1000) - 0.727 \; . \tag{13}$$

where the factor of 1000 is needed because $K_{\text{oc}}$ and $K_{\text{ow}}$ are expressed in m$^3$ kg$^{-1}$ whereas, in the original regression, they used mL g$^{-1}$.

Once $Q$ is computed, the dimensionless fraction of the chemical in the gas phase $F_{\text{gas}}$ is then calculated as

$$F_{\text{gas}} = \frac{\psi}{Q} \; . \tag{14}$$

In the Smit scheme, an empirical relation was established between $F_{\text{gas}}$ and cumulative volatilization ($CV$ in % of substance deposit). $CV$ was determined based on field and greenhouse experiments with numerous pesticides at 21 days after application. For normal to moist field conditions, $CV$ is expressed as

$$CV = 71.9 + 11.6 \log(100\ F_{\text{gas}}) \ ;\ 6.33 \times 10^{-9} < F_{\text{gas}} \leq 1\ , \tag{15}$$

and for dry field conditions,

$$CV = 42.3 + 9.0 \log(100\ F_{\text{gas}}) \ ;\ 0.2 \times 10^{-6} < F_{\text{gas}} \leq 1\ . \tag{16}$$

**Vegetation**

Smit et al. (1998) derived an equation for the cumulative volatilization $CV$ from plants against vapor pressure $P_v$ (mPa) at seven days after application based on field and climate chamber experiments of pesticide volatilization (Equation 17).

$$CV = 10^{1.528 + 0.466 \log P_v}; P_v \leq 10.3\ . \tag{17}$$

For compounds with $P_v$ above 10.3 mPa, $CV$ is set at 100 % of deposit. Temperature adjustments were made for $P_v$ using the Clausius−Clapeyron equation:

$$\frac{d(\ln P_v)}{dT} = -\frac{\Delta H_{\text{vap}}}{R\ T^2}\ . \tag{18}$$

**Snow and glaciers**

The parameterization of substance loss by volatilization from snow pack follows Wania (1997) whereby the process is calculated using a consecutive cycle of an equilibrium partitioning among four phases followed by a contaminant loss. The four phases considered are liquid water, organic matter contained in the snowpack, snow pores (air), and an ice−air interface. Fugacity capacity factors for these phases are expressed with the following relations:

| | | |
|---|---|---|
| air (mol m$^{-3}$ Pa$^{-1}$) | $Z_a = 1/RT$ | (19a) |
| water (mol m$^{-3}$ Pa$^{-1}$) | $Z_l = K_{\text{wa}}/RT = K_{\text{wa}}\ Z_a$ | (19b) |
| organic carbon (mol m$^{-3}$ Pa$^{-1}$) | $Z_o = Z_l\ 0.41\ K'_{\text{ow}}$ | (19c) |
| ice–air interface (mol m$^{-2}$ Pa$^{-1}$) | $z_i = K_{\text{ia}}/RT = K_{\text{ia}}\ Z_a$ | (19d) |

where $R$ is the dry air gas constant (8.312 J mol$^{-1}$ K$^{-1}$), $T$ is the air temperature (K), $K_{\text{wa}}$ is the water−air partition coefficient (unitless), $K'_{\text{ow}}$ is the dimensionless octanol−water partition coefficient, and $K_{\text{ia}}$ is the ice surface−air partition coefficient (m). $K_{\text{ia}}$ at 20°C is estimated using $K_{\text{wa}}$ and water solubility $C^s_w$ (mol m$^{-3}$),

$$\log K_{\mathrm{ia}}\,(20^{\circ}\mathrm{C}) = -0.769 \log C_w^s - 5.97 + \log K_{\mathrm{wa}}\;, \tag{20}$$

and further extrapolated to other temperatures using enthalpy of condensation of solid ($\Delta H_{\mathrm{subl}}$ in J mol$^{-1}$),

$$\log K_{\mathrm{ia}}(T) = \log K_{\mathrm{ia}}\,(20^{\circ}\mathrm{C}) + \frac{0.878\;\Delta H_{\mathrm{subl}}}{2.303\;R}\left(\frac{1}{T} - \frac{1}{293}\right)\;. \tag{21}$$

An equilibrium fugacity $f_s$ is thereby determined by

$$f_s = \frac{M_{\mathrm{sp}}}{Z_a\;v_a + Z_l\;v_l + z_i\;A_{\mathrm{snow}}\;\rho_{\mathrm{mw}} + Z_o v_o}\;, \tag{22}$$

where $M_{\mathrm{sp}}$ is the amount of chemical contained in snowpack (the model here applies snow burden of the chemical in kg m$^{-2}$), $v_a$, $v_l$, and $v_o$ is the volume fraction of air, liquid water, and organic matter in snowpack (m$^3$ m$^{-3}$). For this study, $v_a$ and $v_l$ values are set to 0.3 and 0.1, respectively, whereas $v_o$ is zero assuming no polluted snow. $A_{\mathrm{snow}}$ is the specific snow surface area (m$^2$ g$^{-1}$). In Daly and Wania (2004), a value of 0.1 m$^2$ g$^{-1}$ for $A_{\mathrm{snow}}$ was used for snow accumulation period and a linear decrease from 0.1 to 0.01 m$^2$ g$^{-1}$ was used during the snowmelt period. In SVOC submodel, a value of 0.025 m$^2$ g$^{-1}$ is adopted for $A_{\mathrm{snow}}$ to represent fairly aged snowpack. $\rho_{\mathrm{mw}}$ is the density of snowmelt water and here is taken as $7{\times}10^5$ g m$^{-3}$.

Volatilization rate (kg m$^{-2}$ s$^{-1}$) is calculated by applying

$$\frac{dM_{\mathrm{sp}}}{dt} = \frac{1}{\frac{1}{U_7\;Z_a} + \frac{1}{U_5\;Z_l + U_6\;Z_a}} \times \frac{f_s}{h_s}\;, \tag{23}$$

where $h_s$ is the snow depth (m), $U_5$ is the snow$-$water phase diffusion mass transfer coefficient (m h$^{-1}$), $U_6$ is the snow$-$air phase diffusion mass transfer coefficient (m h$^{-1}$), and $U_7$ is the snow$-$air boundary layer mass transfer coefficient (m h$^{-1}$). $U_5$ and $U_6$ are calculated from molecular diffusivities in air and water (Equations 24 and 25), whereas a typical value of 5 m h$^{-1}$ is adopted for $U_7$.

$$U_5 = B_w\;\frac{v_l^{10/3}/(v_a + v_l)^2}{\ln\;2h_s}\;, \tag{24}$$

$$U_6 = B_a\;\frac{v_a^{10/3}/(v_a + v_l)^2}{\ln\;2h_s}\;, \tag{25}$$

where $B_w$ and $B_a$ are the molecular diffusivities (m$^2$ h$^{-1}$) in water and air respectively. In the model, $B_a$ is derived from the molecular weight (MW) as $B_a = \frac{1.55}{\mathrm{MW}^{0.65}}$ cm$^2$s$^{-1}$, whereas $B_w$ is set as $1{\times}10^4$ less than $B_a$, following Schwarzenbach et al. (2005).

**Ocean**

In the study, the ocean is represented as a surface mixed layer of a depth varying spatially and in time without lateral transports. The mixed layer depths were obtained from (de Boyer Montégut et al., 2004). The SOCś volatilization flux from the sea surface is parameterized based on the two-film model of Liss and Slater (1974) and is calculated within the AIRSEA submodel (Pozzer et al., 2006). Note that no ocean and sea-ice dynamics were
included in the simulations.

Sorption of SOCs in water to suspended particulate matter (colloidal or sinking detritus) is neglected. Therefore, SOC concentration in surface seawater and, hence, volatilization from sea surface is overestimated, in particular for very lipophilic ($\log K_{\mathrm{ow}} > 6$) substances. This bias is negligible for the substances studied here (PAHs) which are less lipophilic or
265 volatilisation is limited by vapor pressure (e.g., benzo(a)pyrene). Forces from strong winds, dissolved or particulate organics in seawater are transferred to air via sea spray, which adds to particulate OM in air over the ocean (O'Dowd et al., 2008; Qureshi et al., 2009). This process is neglected in the model.

### 2.2.4 Dry deposition

Dry deposition is simulated using deposition velocities. For gas-phase SOCs, the velocities are calculated by the DDEP submodel (Kerkweg et al., 2006a), whereas particulate-bound SOCs are assumed to deposit at similar rates to other aerosols whose velocities are also computed by DDEP. If the *modal* scheme is selected (see Section 2.2.1), the particle deposition velocity $v_d^{\mathrm{SOC}}$ at mode $i$ is equal to the aerosol deposition velocity $v_d^{\mathrm{aer}}$ at the respective mode. On the
other hand, for the *bulk* scheme, $v_d^{\mathrm{SOC}}$ is computed as a weighted average of $v_d^{\mathrm{aer}}$ from the four BC modes (*ki*, *ks*, *as*, and *cs*) where the weight is the surface area of BC. This approach is most relevant for PAHs as they are assumed to be predominantly transported by sorption to BC. The above relations are formulated as follows

$$\textit{modal scheme:} \qquad v_{d,i}^{\mathrm{SOC}} = v_{d,i}^{\mathrm{aer}} \tag{26a}$$

$$\textit{bulk scheme:} \qquad v_{d,\mathrm{bulk}}^{\mathrm{SOC}} = \frac{\sum\limits_{i=1}^{4} S_{\mathrm{BC}_i} \times v_{d,i}^{\mathrm{aer}}}{\sum\limits_{i=1}^{4} S_{\mathrm{BC}_i}} \tag{26b}$$

and the BC surface area per unit volume $S_{\mathrm{BC}}$ (cm$^2$ cm$^{-3}$) is given by

$$S_{\mathrm{BC}_i} = 4\pi \left[ r_i \exp\left(\ln^2 \sigma_{g_i}\right) \right]^2 N_i \times \frac{C_{\mathrm{BC}_i}}{C_{\mathrm{aer}_i}} \ , \tag{27}$$

where $N_i$ is the number concentration for mode $i$ (cm$^{-3}$), $r_i$ is the number radius (cm), $\sigma_{g_i}$ is the geometric standard deviation, $C_{\mathrm{BC}_i}$ is the BC concentration ($\mu$g m$^{-3}$) in mode $i$, and $C_{\mathrm{aer}_i}$ is the sum of aerosol concentrations in the same mode ($\mu$g m$^{-3}$).

### 2.2.5   Wet deposition

Wet deposition is applied to both gas and particulate SOCs. The gaseous fraction is scavenged into cloud and rain droplets according to diffusion limitation, Henry's law equilibrium, and accommodation coefficient, and this process is parameterized and solved empirically in the SCAV submodel (Tost et al., 2006a). Particulate-phase SOCs are scavenged in convective updrafts, rainout and washout, and cloud evaporation, with the rate being proportional to BC wet scavenging; hence the change in SOC concentration is described as

$$modal \text{ scheme:} \qquad \frac{\Delta C_{\text{SOC}_i}}{\Delta t} = \frac{\mu_{\text{SOC}_i}}{\mu_{\text{BC}_i}} \times \frac{\Delta C_{\text{BC}_i}}{\Delta t} \tag{28a}$$

$$bulk \text{ scheme:} \qquad \frac{\Delta C_{\text{SOC}}}{\Delta t} = \frac{\mu_{\text{SOC}}}{\sum\limits_{i=1}^{4} \mu_{\text{BC}_i}} \times \sum\limits_{i=1}^{4} \frac{\Delta C_{\text{BC}_i}}{\Delta t} \tag{28b}$$

where $\mu$ is the particle volume mixing ratio ($\text{mol}_{\text{SOC/BC}} \text{ mol}_{\text{air}}^{-1}$) and $\Delta t$ is the model time step (s). Note that Equation 28a imposes a restrictive prerequisite, namely, BC and the particle-bound SOC have similar size distributions. When this condition is not met, there is a high possibility of an artificial mass being produced, usually to the largest aerosol mode. To solve this problem, a correction factor is applied and defined as a function of the ratio of positive fluxes to negative fluxes integrated across levels and modes.

### 2.2.6   Atmospheric degradation

The atmospheric degradation of SOCs in the gas phase as well as within aerosol particles are explicitly treated in SVOC. The gas-phase chemical mechanism is calculated within the MECCA submodel (Sander et al., 2011). SOC gaseous degradation is from photochemical reactions with OH, $NO_3$, and $O_3$ radicals which follow a $2^{\text{nd}}$-order transformation, with the rate constants $k^{(2)}$ obtained from laboratory studies. $k^{(2)}_{\text{OH}}$ value is typically higher, suggesting that oxidation with OH radical is the dominant loss pathway.

Most models do not consider oxidation rate of particulate-phase SOCs as experimental aerosols studied in laboratory cover only a small part of atmospheric relevant aerosols. For PAHs, such as benzo(a)pyrene which stays mostly in the particulate phase, the degradation is more efficient by surface reactions with $O_3$ (Shiraiwa et al., 2009, and references therein) with the rate depending on the substrate. The SVOC submodel includes degradation process of PAHs on aerosol particles from the $O_3$ reaction with one assumption, that is, the heterogeneous reaction does not lead to a change in the oxidant concentration. Due to a limited number of kinetic studies of heterogeneous reactions of PAHs, only two species are considered (phenanthrene and benzo(a)pyrene). Nevertheless, the submodel structure provides a relatively straightforward approach to allow more species in the future.

The reaction rate coefficient for particulate-phase PHE with $O_3$ at aerosol surfaces was derived from laboratory experiments using chemically unspecific model aerosol (silica) with PAH surface coverage of less than a monolayer (Perraudin et al., 2007). To this end, the

second order rate coefficient, $k^{(2)}$, in cm$^4$ molec$^{-1}$ s$^{-1}$ was derived from the reported PHE decay kinetics, $k^{(2)}_{\mathrm{O_3,het}}$ (cm$^3$ molec$^{-1}$ s$^{-1}$), as $k^{(2)} = k^{(2)}_{\mathrm{O_3,het}}/\left(\frac{S}{V}\right) = (6.2 \pm 4.8) \times 10^{-17}$ cm$^4$ molec$^{-1}$ s$^{-1}$, with $\frac{S}{V}$ (cm$^{-1}$) being the experimental aerosol surface concentration ($0.56 \pm 0.43$ cm$^{-1}$ in Perraudin et al. (2007)). In the submodel, $k^{(2)}_{\mathrm{O_3,het}}$ (cm$^3$ molec$^{-1}$ s$^{-1}$) is calculated using the ambient aerosol surface concentration. As for BaP, the pseudo-first-order rate coefficient, $k^{(1)}_{\mathrm{O_3,het}}$ in s$^{-1}$, was derived from surface-adsorbed BaP reaction with O$_3$ on solid organic and salt aerosols following the Langmuir−Hinshelwood mechanism (Kwamena et al., 2004).

### 2.2.7 Biotic and abiotic degradations

Biotic and abiotic processes in surface compartments contribute to the degradation of chemicals and are strongly dependent on local environmental conditions, for example, nutrient contents, water, temperature, PH, and light. In SVOC, these factors are not explicitly quantified. The degradation is alternatively described as following a first-order decay law (Equation 29). The 10 K temperature warming is assumed to double the rate of degradation (Equation 30), following recommendations in chemicals risk assessment (European Commission, 2000) and consistent with findings, such as a two-time increase in the growth of hydrocarbon-degrading microbes found in soils (Thibault and Elliott, 1979).

$$\frac{\partial C_{\mathrm{SOC_s}}}{\partial t} = -k_{\mathrm{sfc}} \times C_{\mathrm{SOC_s}} \; , \tag{29}$$

$$k_{\mathrm{sfc}(T)} = k_{\mathrm{sfc}(T_{\mathrm{ref}})} \times 2^{\frac{T - T_{\mathrm{ref}}}{10}} \; , \tag{30}$$

where $C_{\mathrm{SOC_s}}$ is the substance concentration (kg m$^{-3}$) in surface compartments (that is, soil, vegetation, or ocean) and $k_{\mathrm{sfc}}$ is the first-order decay rate (s$^{-1}$). $T_{\mathrm{ref}}$ is the reference temperature, that is, 298 K for soil and 273 K for ocean. Note that the degradation in vegetation is calculated assuming the same $k_{\mathrm{sfc}}$ for the soil compartment.

## 2.3 Input data

### 2.3.1 Kinetic and physicochemical properties

The model simulations were performed for four PAH species: phenanthrene (PHE), pyrene (PYR), fluoranthene (FLT), and benzo(a)pyrene (BaP). To simulate the fate and environmental distribution of these species, the model requires some physicochemical properties as summarized in Table S5 of the supplement. These include equilibrium partition coefficients and their related energies of phase transfer. The characteristics from PHE to BaP are indicated by decreasing volatility (as molar mass increases), increasing $K_{\mathrm{oa}}$ and $K_{\mathrm{ow}}$, and decreasing water solubility (as $C_w^s$ and Henry's coefficients decrease). The properties also include the second-rate coefficients for homogeneous oxidation with OH, O$_3$, and NO$_3$ except for BaP where the gaseous reaction is switched off. Heterogeneous oxidation by O$_3$ is simulated only

for PHE and BaP. Furthermore, the model also requires compound solute parameters for simulations using the ppLFER gas−particle partitioning scheme (Table S6).

### 2.3.2 Emissions and other model input

As model input, several emission datasets were employed in the study. Emission estimates for PAHs were obtained from the annual mean inventory of Shen et al. (2013) for the year 2008. They applied regression and technology split methods to construct country-level emissions for six categories (coal, petroleum, natural gas, solid wastes, biomass, and an industrial process category) or six sectors (energy production, industry, transportation, commercial/residential sources, agriculture, and deforestation/wildfire) before further regridding the emissions to a $0.1° \times 0.1°$ grid.

Emissions of aerosol species such as organic carbon (OC), black carbon (BC), mineral dust (DU), and sea salt (SS) were included. For BC and OC, the Representative Concentration Pathway (RCP) 6.0 emission scenario of the IPCC (Intergovernmental Panel on Climate Change) (van Vuuren et al., 2011) was used and accessible via ftp://ftp-ipcc.fz-juelich.de/pub/emissions/gridded_netcdf. Emissions are calculated for anthropogenic, biomass burning, ship, and aircraft. The RCP database provides a seasonality only for the biomass burning and ship emissions. In this study, seasonal scale factors were applied to the anthropogenic emission whereby the seasonality was based upon the monthly variation of the Hemispheric Transport of Air Pollutants (HTAP) v2.2 anthropogenic emission inventory (Janssens-Maenhout et al., 2015). BC emissions from all sectors were assumed to be hydrophobic. For OC, it was assumed to be 65% hydrophilic and 35% hydrophobic upon biomass burning emissions and to be 100% hydrophobic upon anthropogenic and ship emissions. Both OC and BC were emitted at Aitken mode which spans the size range from about 5 to 50 nanometer in diameter. A factor of 1.724 was used to scale the OC emissions to primary organic matter (OM). It is noteworthy that the formation of secondary organic aerosols (SOA) from atmospheric oxidation and condensation of volatile organic compounds (VOCs) were not treated in the simulations. In the model, particulate organic matter is emitted and transported as a bulk aerosol species (OM).

DU and SS emissions were computed online by the ONEMIS submodel (Kerkweg et al., 2006b). DU emission flux is calculated based on wind speed at 10 m altitude and soil parameters (Schulz et al., 1998). The emission of SS particles by bubble bursting is described as wind-speed dependent particle mass and number fluxes at accumulation (50−500 nm) and coarse (>500 nm) modes. In ONEMIS, the fluxes are determined from pre-calculated lookup tables following the Guelle et al. (2001) parameterization. SVOC submodel accounts for OM fraction in the SS mass fluxes ($J_{\text{SS}}$), and the fraction is estimated using a 10 m wind ($v_{10}$)-dependent empirical relationship derived from Figure 2a in Gantt et al. (2011). Equation 31 below is used to calculate the OM mass fluxes, $J_{\text{OM}}$, in kg m$^{-2}$ s$^{-1}$.

$$J_{\text{OM}} = J_{\text{SS}} \times \frac{1}{2} \left( \frac{0.78}{1 + 0.03 \exp(0.48 v_{10})} + \frac{0.24}{1 + 0.05 \exp(0.38 v_{10})} \right) . \tag{31}$$

Emissions of other gases including volatile organic species ($SO_2$, CO, $NH_3$, NO, $CH_4$, and NMHC) were prescribed using the IPCC RCP6.0 dataset (van Vuuren et al., 2011). Global estimates for the soil properties, dry bulk density and organic matter fraction, were obtained from Dunne and Willmott (1996) and Batjes (1996), respectively.

## 2.4    Observational data

The observation data used for model performance evaluation were collected from several surface monitoring networks: the European Monitoring and Evaluation Programme (EMEP) (Tørseth et al., 2012), the Arctic Monitoring and Assessment Program (AMAP) (Hung et al., 2005), the Great Lakes Integrated Atmospheric Deposition Network (IADN) (IADN, 2014), the Department for Environment, Food & Rural Affairs (DEFRA) UK-AIR (Air Information 395    Resource) program (DEFRA, 2010), and the MONitoring NETwork in the African continent (MONET-Africa) (Klánová et al., 2008). These data were screened and quality controlled according to the description in Supplement SIII. Final stations with reliable monthly data are depicted in Figure 2 wherein shows 3 stations in the Arctic, 19 in the northern mid-latitudes, and 6 in the tropics. The availability of data differs by station, species, and variable of 400    interest; see the site-specific information in Table S10 for total concentration and Table S11 for particulate mass fraction ($\theta$).

Insert Figure 2

The study also compared simulated concentrations in the marine atmosphere to two ship 405    cruise measurement campaigns: 1) on a West to East transect across the tropical Atlantic Ocean (Lohmann et al., 2013), and 2) along the Asian marginal seas, the Indian, and the Pacific Oceans (Liu et al., 2014). The monthly mean modeled values were compared to daily measurements at each sampling points.

## 2.5    Experiment designs

### 2.5.1    Model configuration

The model was run on a spectral T42 grid in the horizontal (approximately $2.8°$ in a lat−lon grid) and 19 unevenly distributed layers in the vertical with the top level at 10 hPa. The vertical layers are discretized using a hybrid coordinate (the lowest level follows the terrain and becomes surfaces of constant pressure in the stratosphere). All simulations were run for a 415    three-year period (i.e., 2007−2009), with a one-year spin-up (i.e., 2006), and nudged toward the European Centre for Medium-range Weather Forecasts (ECMWF) reanalysis data (Dee et al., 2011). Note that the simulation period was selected based on the representative year of

PAH emissions (i.e., 2008) and the availability of reliable observation data (see Supplement SIII).

### 2.5.2 Sensitivity to the temporal resolution of emissions and process parameterizations

Factor separation analysis (Stein and Alpert, 1993) was used to quantitatively evaluate the contributions to changes in a particular output variable that result from changing components of model input and physics parameterizations. The model sensitivity to four model components (hereinafter, "factors") was tested. The four factors were:

1. Temporal resolution of emissions (hereinafter, *fac1*)
   The PAH emission inventory of Shen et al. (2013) was based on 2008 annual emission totals from all sectors (see Section 2.3.2). The emissions were divided over the year using monthly factors derived from BC anthropogenic emissions. Two sets of simulations were carried out to test the sensitivity of model output to the seasonal profile of emission. The first set used constant emissions throughout the simulations whereas the second set used monthly emission interval.

2. The size-discretization of particulate-phase PAHs (hereinafter, *fac2*)
   The two options for this factor were tested: *bulk* versus *modal* (Section 2.2.1). Note that with regards to BaP, 95% of the emissions were assumed to be in particulate phase and for the *modal*-scheme scenario, all of the emitted particles are treated as the hydrophobic Aitken ($ki$) tracers.

3. The choice of gas−particle partitioning scheme (hereinafter, *fac3*)
   The present study focuses on the comparison between the Lohmann−Lammel and ppLFER schemes for gas−particle partitioning.

4. The influence of re-volatilization (hereinafter, *fac4*)
   Model runs with volatilization process switched off are compared to those runs which have volatilization switched on.

The factor separation technique is described in Supplement SIV including the equations used to compute the model sensitivity to four factors. A total of 16 (or $2^n$) experiments summarized in Figure 3 are necessary to supply the complete solution for the factor analysis. The ABLN experiment is designed to be the *base* simulation ($f_0$), in which annual emission (A), the *bulk* scheme (B), the Lohmann−Lammel scheme (L), and no re-volatilization (N) were applied. SMPW is referred to as the *target* simulation ($f_{1234}$) in which the more sophisticated choice of the four features (factors) were tested, i.e., seasonal emissions + *modal* scheme + ppLFER scheme + with re-volatilization. The total (gas + particle) concentration at the lowest model level was selected for its higher relevance with all the factors (compared to, e.g., atmospheric burden) and to facilitate direct comparison with observations.

# 3 Results and discussion

## 3.1 Sensitivity tests

The analysis of the factor separation results is given below. For each factor, the analysis includes the assessment of direct effects ($\hat{f}_i$) and total interaction effects ($\Sigma\hat{f}_{ij} + \Sigma\hat{f}_{ijk} + \hat{f}_{1234}$) on near-surface PAH concentrations in two seasons, i.e., December−January−February (DJF) and June−July−August (JJA). Figure 4 shows the respective effects for all factors as relative to the seasonal means of the *base* experiment ($f_0$). A positive value indicates a concentration increase with respect to $f_0$, whereas a negative indicates a decrease. The spatial distributions of $f_0$ and $f_{1234}$ seasonal mean concentrations for the four species are shown in Supplement SVI, Figures S4 and S5 respectively.

We studied the relative effects in five climate zones (Arctic, northern mid-latitudes, Tropics, southern mid-latitudes, and Antarctica) The global distributions of the relative effects are presented in Figures S6−S13 whereas Figures S14−S21 present the relative interaction effects from the individual combination of factors. In the following, we do not look to interpret concentration responses to each interaction term. Reasons for this are that (1) accounting for all such interactions is complicated given the number of factors and (2) higher-order interactions (combinations of more than two factors) is hard to physically interpret.

We further investigate the factor effects on model performance by comparing the predicted seasonal mean near-surface concentrations from 16 experiments against observation data in the Arctic and northern mid-latitudes (Supplement SVII).

### 3.1.1 Effects of seasonality of emissions

Figure 4a shows that using monthly emissions increases PAH concentrations in DJF and decreases the concentrations in JJA over the areas from the middle to high latitudes of NH. This result is expected and is attributed to emissions during the northern winter (summer) being higher (lower) than annual means and photochemistry being less (more) active. Over the Arctic, the relative changes ($\hat{f}_1/f_0$) in DJF show a median increase of 30% for PHE, PYR and FLT, and 7% for BaP, whereas $\hat{f}_1/f_0$ in JJA are weaker in magnitude for PHE and PYR ($-16\%$) but comparable for FLT ($-28\%$) and BaP ($-5\%$). Accounting for seasonality leads to generally lower bias for PHE, and this effect is more pronounced in middle than in high latitudes (Supplement SVII, Figure S22).

In general, $\hat{f}_1/f_0$ becomes smaller over the northern mid-latitudes by around half. The upper (lower) quartile of $\hat{f}_1/f_0$ in DJF (JJA) indicates about one-quarter areas of the temperate and polar regions experience at least 40% of an increase (decrease), most were located in northeastern Eurasia (see the left panels of Figures S6 and S7). Note $\hat{f}_1/f_0$ over the tropics are small-to-negligible ($\pm1\%$) mainly due to little variation in emissions from anthropogenic sectors. PAH concentrations may be higher in dry season due to increased amounts of biomass burning, but they are poorly represented in the current inventory. In southern mid- and high-latitudes, the direct effects of emission change are substantially opposite in sign to the

effects seen in the northern latitudes, being negative in DJF (median ranges from $-4\%$ to $-32\%$) and positive in JJA ($7\%-25\%$).

Insert Figure 4

The total interactions between *fac1* and other factors generally produce opposite signals to $\hat{f}_1$ over middle and high latitudes in the two seasons. This result indicates that the changes in other factors tend to buffer the influence of monthly emission on increasing or decreasing $f_0$ concentrations. Some exceptions are seen over parts of East Asia in DJF for all species (Figure S6, right panels) and over the Southern Ocean in JJA for BaP (Figure S7, right panels) where

the interactions work to reinforce the direct effects. In DJF, the degree of interactions is smaller or comparable to the size of $\hat{f}_1$ for the Arctic and northern mid-latitudes but becomes stronger by at least double for Antarctica. The opposite tendency is seen in JJA but only applies to PYR and FLT. In agreement to $\hat{f}_1$, the interaction effects are less apparent over the tropics. Note that the positive effects in $\hat{f}_{14}$ during local summer tend to be more dominant

than the effects in other combinations for PHE, PYR, and FLT (Figures S18−S20). In the simulation, the presence of re-volatilization in summer tends to suppress $\hat{f}_1$ by promoting more gases available for long-range transport, thus implies a negative feedback.

### 3.1.2   Effects of size-discretization of particulate-phase tracer

The direct effects of the *modal* scheme ($\hat{f}_2$) vary among species (Figure 4c). $\hat{f}_2$ is almost

510 absent for PHE as the species resides almost completely in the gas phase. For PYR and FLT, $\hat{f}_2$ is negative during DJF over northern mid-latitudes ($\hat{f}_2/f_0$ quartiles range from $-5\%$ to $-30\%$) and the Arctic ($-50\%$ to $-75\%$) whereas it is hardly visible in JJA or over other regions. Further analysis reveals stronger particle deposition results when the aerosol phase is discretized into different modes (not shown). In long-range transport under modal

aerosol representation, the aerosols are more associated with larger particles hence particle deposition becomes more effective. The choice of size discretization has only minor effects for atmospheric levels, except for BaP, especially during DJF, for which overestimates are significantly compensated for (Figure S25). Actually, for BaP, the *modal* scheme generally decreases the concentrations in the Arctic (as median, $-35\%$ in DJF and $-15\%$ during JJA)

and increases (approx. 5%) those over mid- and low-latitude landmass (Figures S8d and S9d, left panels).

    As is the case for the direct effects, the interaction contributions are peculiar to individual species (Figure 4d). For PHE, the interaction effects in DJF are reflected in negative concentration responses over the Arctic (18%) and positive over Antarctica (7%), in contrast

to relatively mild influences over other regions or in JJA. For PYR and FLT, the effects are negative over the Arctic both in DJF (quartiles vary from $-20\%$ to $-75\%$) and in JJA ($-6\%$ to $-120\%$). It is interesting to note that the interaction between the *modal* and ppLFER schemes has a major influence on the negative signal (Figures S15, S16, S19, S20), suggesting that the

decrease in simulated concentration associated with the change from *bulk* to *modal* could be intensified when the ppLFER scheme is used. In the remaining areas, the interaction effects vary in sign spatially as illustrated in the right panels of Figures S8b,c−S9b,c. Nevertheless, it shows for both species that maximum influences occur over the Southern Ocean in DJF (where the effects may reach two orders of magnitude) and mid-latitude landmass in JJA (more than a factor of five). As for BaP, the median effects are negative ($-7\%$ to $-30\%$) in both seasons, although some positive signals are apparent in parts of high latitudes while the tropical oceans bear small synergistic effects. Similar to other species, the degree of interactions are stronger than $\hat{f}_2$ by more than a factor of three for the majority of grid cells (Figures S8d−S9d, right panels). The large fractions of the effects are dominated by two-factor and three-factor combinations related to the interaction with the ppLFER scheme and/or re-volatilization (Figures S17, S21).

### 3.1.3 Effects of the choice of gas−particle partitioning scheme

Figure 4e shows that the direct effects of the ppLFER scheme ($\hat{f}_3$) show little spatial heterogeneities in both seasons and for all species. The effects are barely important for PHE due to low gas−aerosol partition constant ($K_p$). $\hat{f}_3$ is positive for PYR and FLT over polar regions and northern mid-latitudes especially in winter when low temperature favors partitioning to aerosols (higher $K_p$). The median of $\hat{f}_3/f_0$ varies from $1\%$ to $25\%$ with some parts of Antarctica showing an increase larger than $50\%$. For BaP, the effects are overall negative (by at least $-5\%$) with $\hat{f}_3/f_0$ reflecting a positive north−south gradient (increasing from the Arctic to Antarctica), associated in part with stronger signals over oceans (Figures S10d and S11d, left panels). In particular under the modal size discretization, the choice of gas−particle partitioning scheme has only minor effects for atmospheric levels, except for BaP for which model overestimates are compensated by the choice of the ppLFER scheme (Figure S25). Under the bulk size discretization, the ppLFER scheme tends to enhance some of the overestimate in the Arctic summer (FLT, PYR; Figures S23−S24). The application of ppLFER increases Kp as this module is calculated from not only interaction with BC and OM (as in Lohmann−Lammel scheme) but also with some other aerosol matrices. Higher $K_p$ indicates higher particle mass fraction. For PYR and FLT, this leads to an increase in total atmospheric lifetime as the aerosol phase is not degraded, and can, therefore, be transported over a larger distance. For BaP, the additional particles are subject to depositions and heterogeneous oxidation by ozone, particularly in regions away from sources. The factor influence is notably too small for PHE as oxidations occur in both phases.

The effects from *fac3* interactions vary by region and are relatively stronger than $\hat{f}_3$ (Figure 4f). This finding is common to all species and seasons. The degree of effects is weaker for PHE compared to that for other species. However, the interactions increase polar concentrations in local summer, by $20\%$ to a factor of five, mainly associated with the coupled effect of ppLFER and volatilization ($\hat{f}_{34}$, Figures S14 and S18). For PYR and FLT, there is a high spatial variability over extratropical regions in local summer, as indicated by the interquartile range (distance between the 3rd and 1st quartiles). With regard to synergistic terms, ppLFER

interactions with the *modal* scheme and re-volatilization, in two- or three-factor combinations, are more important than other contributions (Figures S15−S16 and S19−S20). For BaP, the interaction effects show negative signals similar to $\hat{f}_3$, suggesting a positive feedback. The interactions exert a stronger influence on the concentrations of the oceans than on that of land, except in the tropics (Figures S10d and S11d, right panels). The median of relative effects ranges from −1% to a factor of −10, minimum (maximum) in the northern (southern) extratropics. Two second-order interactions likely make major contributions, that is, $\hat{f}_{34}$ which dominates the response over oceans, and $\hat{f}_{23}$ which dominates over land (Figure S17 and S21).

### 3.1.4 Effects of re-volatilization

The direct effects of re-volatilization ($\hat{f}_4$) are illustrated in Figure 4g. $\hat{f}_4$ is positive in the tropics in both seasons, with the median $\hat{f}_4/f_0$ ranging from 5% to 50%. Intensive re-volatilization in this region would increase net surface fluxes, thereby increasing concentrations. For PHE, positive $\hat{f}_4$ values are more localized over the tropical landmass, whereas negative $\hat{f}_4$ values are predicted over the tropical ocean (Figures S12a and S13a; left panels). The positive (negative) effects over land (ocean) areas are also apparent at higher latitudes during most of the year. This reflects the fact that the negative effects on concentrations over ocean act contrary to the positive effects on net surface fluxes, mainly caused by the non-linear relationships of air−sea gas exchange (deposition and volatilization), air and surface burden, atmospheric oxidation, and emissions. Accounting for re-volatilization compensates for a significant part of underestimates of PHE in the Arctic during summer, but adds to overestimates in mid-latitudes (Figure S22).

For the studied species of mid semivolatility, PYR and FLT, a positive signal is apparent over the high and middle latitudes during local summer in contrast to a negative signal during local winter (Figure 4g). Similar to PHE, the negative signal is confined over oceans (Figures S12b−c and S13b−c; left panels). The summer increases are stronger (20% to a factor of ten) than the winter decreases (−10% to −60%) and the magnitudes are higher in FLT than in PYR. The near-ground concentrations of PYR and FLT are estimated by ≈30−80% in mid-latitudes of which ≈30% are attributable to re-volatilization. In the Arctic, re-volatilization compensates for ≈60% of PYR underestimation (Figure S23) and explains most of ≈60−80% of FLT overestimation (Figure S24). For BaP, $\hat{f}_4$ is positive consistently across regions and seasons ($\hat{f}_4/f_0$ ranges from 20% to a factor of ten), with substantial effects occurring over oceans (Figures S12d−S13d; left panels). Accounting for re-volatilization creates some overestimates in the Arctic during summer (Figure S25). It should be noted that the parameterization adopted here to describe volatilization from soils (the Smit scheme) is derived from an experimental study on mid-polar to polar pesticides and there is a need to validate and eventually sophisticate the parameterization to apolar substances.

The interactions generally point toward positive effects for the high-to-medium volatility species (Figure 4h), despite some negative effects present over parts of the southern (northern) oceans in DJF (JJA) (Figures S12−S13; right panels). As for BaP, the effects are uniformly negative, inferring the interactions work in opposition

to $\hat{f}_4$. The negative response is almost entirely caused by the negative $\hat{f}_{34}$, that is, the two-factor interaction between re-volatilization and the ppLFER scheme (Figures S17,S21). Compared to $\hat{f}_4$, the degree of interactions are weaker for PHE, except in polar regions during local summer where the interactions could amplify $\hat{f}_4$. The above implies that $\hat{f}_4$ may point in the right direction regardless of the influences from other factor changes. In contrast, the degree of interactions is overall comparable to $\hat{f}_4$ for the other species.

## 3.2 Model evaluation

Model performance using the sophisticated realization of the four features (factors), i.e., Seasonal emission + *Modal* scheme + ppLFER scheme + With re-volatilization (SMPW) is presented below. Two predicted variables are evaluated, that is, total (gas+particle) concentrations and aerosol particulate mass fraction at the lowest model level. The metrics applied is listed in Supplement SV.

### 3.2.1 Near-surface air concentration

#### Comparison to land monitoring stations
*Central tendency.* Table 2 shows statistical indices for near-surface concentrations of atmospheric PAHs from observations and simulations and their comparisons, averaged across stations in the Arctic, northern mid-latitudes, and the tropics. We can see that mean observed concentrations are higher for PHE and smaller for BaP over all regions. Furthermore, the Arctic concentrations are lower than those in the northern mid-latitudes by a factor of around 20 and those in the tropics by approx. two orders of magnitude. The model captures well these species and regional variations, but the magnitudes are both under- and over-estimated. In the Arctic, it underestimates PHE (MB = $-0.060$ ng m$^{-3}$) and BaP (MB = $-0.006$ ng m$^{-3}$) concentrations but slightly overestimates PYR (MB = $0.001$ ng m$^{-3}$) and FLT (MB = $0.04$ ng m$^{-3}$). In the NH mid-latitudes, the model overestimates the three species predominantly occurring in the gas phase (MB = $0.077-0.867$ ng m$^{-3}$) but underestimates BaP (MB = $-0.58$ ng m$^{-3}$). Negative bias is seen in the tropics for three PAHs (MB = $-3.443$ to $-6.851$ ng m$^{-3}$). Nevertheless, the comparison of model and observations at individual monitoring station can be different from the regional mean statistics, as described in Supplement SVIII. Comparing all four PAHs, a larger degree of bias is found for BaP which increases from the northern mid-latitudes (NMB = $-0.58$, NMBF = $-1.40$, FAC2 = $0.31$, FAC10 = $0.79$) to the Arctic (NMB = $-0.92$, NMBF = $-12.17$, FAC2 = $0.17$, FAC10 = $0.33$).

Insert Table 2

*Dispersion of monthly concentrations.* In the following, the coefficient of variation (CoV) is used to compare the dispersion of concentrations among species of different ranges. CoV

was calculated by dividing standard deviation (SD) of all data points by its mean value
($\overline{x}$). The observations show high variability (CoV > 1) with CoV ranging between 1.12 and
2.14. The simulated concentrations appear to be less dispersed than the observations (CoV =
0.78−1.93) except for the Arctic PHE and PYR concentrations. The degree of underestimation
is larger in the tropics with CoV being 30%−50% smaller than the observations. Furthermore,
correlations between predicted and observed concentrations are weaker than those in other
regions where $r$ varies between 0.29−0.63 (the model reproduces 8%−40% of the variance
in observed concentrations). Comparing the four species, the simulated BaP shows greater
underpredictions of the variability where CoV values are less than half of those observed and
correlations are less than 0.2 (accounting not over than 4% of observed variance). Higher
variability in BaP measurements (than in model results) can be influenced by strongly varying
emissions in source regions that are not reflected in emission inventory (Matthias et al., 2009).

*Seasonal variation.* Figure 5 compares simulated and observed seasonal cycle of average
concentrations for different species and regions. The observed mean concentrations are largest
in winter and lowest during summer because of less emission and the strong presence of
OH for oxidation. The winter maximum to summer minimum ratio (amplitude) is more
pronounced (by more than a factor of two) in the Arctic than that in the NH mid-latitudes.
The seasonality between model and observations is in qualitative agreement, particularly over
the Arctic (except in summer) and mid-latitudes. In the Arctic, the model overestimates the
seasonal amplitude of PHE and BaP and underestimates their mean concentrations. The
contrast is seen for PYR and FLT. FLT concentration is overestimated by up to a factor
of three in summer while PYR is quite well predicted. In the NH mid-latitudes, the model
underestimates the amplitude but overestimates the concentrations of PHE, PYR, and FLT
(by typically a factor of two), whereas a systematic negative bias is found for BaP. In the
tropics, both the amplitude and magnitude are too low in the model (for magnitude, by a
factor of 2−5).

Additional findings are discussed in Supplement SIX related to the comparison between EMAC
model results and those from other global PAH modeling studies.

Insert Figure 5

**Comparison to ship cruise measurements**

Measurements of PHE, PYR, and FLT concentrations over the Atlantic Ocean were taken
during a cruise in July 2009 (Lohmann et al., 2013). Figure 6 shows the ship sample
concentrations overlaying the simulated PAH concentrations. Sample arithmetic (geometric)
means during the whole cruise transect are 322 (209), 95 (88), and 128 (111) pg m$^{-3}$ for PHE,
PYR, and FLT respectively. The model poorly reproduces the remote marine environments
and overall underestimates the observations, except at 3 locations along the North American
coast. The simulated means across sampling positions are 23 (7), 20 (3), and 39 (2) pg m$^{-3}$
respectively and the underestimation ranges from a factor of 2 to 1000. The degree of bias is
most apparent over the tropical South Atlantic at latitude bands 5°S-15°S.

As reported in Liu et al. (2014), the measured concentrations of BaP over the Asian marginal seas, the Indian Ocean, the South and North Pacific Oceans are 131 (45), 14 (3), 9 (2), and 8 (3) pg m$^{-3}$, respectively, for the arithmetic (geometric) means of all samples. Similar to other species, the model also underestimates the BaP concentrations with mean values being 75 (15), 4 (0.05), 0.09 (0.03), and 0.2 (0.06) pg m$^{-3}$, respectively. The discrepancy appears relatively smaller over the Asian marginal seas as compared to other locations (Figure 7). A substantial degree of bias is seen over the Indian Ocean covering approximately the area bounded by 70°E−90°E and 10°S−30°S, with simulated values being more than two orders of magnitude smaller than the observed.

Insert Figure 6
Insert Figure 7

The model tendency to underestimate the marine air concentrations may likely be due to several factors: (a) The grid resolution is not sufficient to reproduce fine-scale processes at the grid points close to shipping tracks; (b) Great uncertainties associated with the air−sea gas exchange parameterizations still exist, most notably in the estimation of gas transfer velocity; (c) The global inventory (Shen et al., 2013) may significantly underestimate emissions from ocean shipping and does not well characterize the spatial and temporal variability of biomass burning plumes as another potential point of origin of pollutans in the marine air (Nizzetto et al., 2008); (d) PAH concentration over remote oceans is controlled by atmospheric components (e.g., temperature, wind speed, boundary layer height, photochemical degradation) and the dynamical and biogeochemical components of the ocean. However, the ocean components have not been covered in the simulation; (e) The particulate-bound PAHs may undergo too fast heterogeneous oxidation (most relevant for BaP), leading to short atmospheric lifetimes and weaker long-range transport. BaP, mostly stays in the particulate phase, presumably also in seawater, therefore, may be somewhat underestimated due to the neglect of sea-spray driven aerosol suspension.

### 3.2.2 Particulate mass fraction

Measurements of particulate mass fraction ($\theta$) were available only from E3 station in Europe and IADN stations (I1-I7) in North America (see Table S11). Table 3 presents summary statistics on monthly mean $\theta$ from observations and simulations including some performance metrics. The observed mean $\theta$ is smaller for PHE ($0.051 \pm 0.035$) and higher for BaP ($0.949 \pm 0.067$). This result is expected as volatility decreases (hence $\theta$ increases) from (lighter) PHE to (heavier) BaP. The $\theta$ values for PYR and FLT are larger by over five times than those for PHE and lower by around one-third than those for BaP. The model reproduces well the distinct differences among species but underestimates the observed $\theta$ for PHE, PYR and FLT. The degree of negative bias is relatively large in PHE (NMB $= -0.910$ and NMBF $= -10.145$), whereas for the isomer pair of PYR and FLT, the model exhibits a similar performance with

a slight improvement in PYR (NMB = −0.410 and NMBF = −0.694). With regard to BaP, there is a satisfactorily small bias (MB = 0.015, RMSE = 0.074, NMB = 0.016, and NMBF = 0.016) although the observed and simulated values have a very weak correlation ($r = 0.03$).

Figure 8 shows the seasonal mean $\theta$ averaged over three years for all PAHs. Observations show that $\theta$ for BaP varies less than those for 3−4 ring PAHs. Although the model adequately reproduces this feature as well as seasonal variation of individual species, the simulated $\theta$ of PHE, PYR, and FLT are generally lower than the observations (except for PYR in winter). For BaP, differences between model and observations are less than 10% in all months. The SVOC submodel describes the gas−particle partitioning of atmospheric SOCs as a function of temperature and aerosol phase composition. The underestimation might be related to the fact that the submodel assumes the particle to be fully in equilibrium with the gas phase at all times. It neglects kinetic limitations of molecular diffusivity that could lead to the trapping of particles inside viscous (or semisolid) organic aerosol coating. This shielding effect increases equilibration times of the particles, thereby reducing part of $\theta$ from the mass available for gas−particle partitioning. Deviations from measurements can also be partly attributed to the locations of some stations that are within, or close to, residential and industrial area (namely, I4, I6, and I7) where the scale and gradient in anthropogenic emissions are not resolved by the model grid resolution nor represented by the emission inventory.

Insert Table 3
Insert Figure 8

# 4   Summary and conclusions

The submodel SVOC has been developed and operated within the EMAC model for the application to global distribution and environmental fate of SOCs. In this first development, the focus was set on the predictions of four PAH species: phenanthrene (PHE), pyrene (PYR), fluoranthene (FLT), and benzo(a)pyrene (BaP). Multicompartmental fate and air−surface exchange processes were included in SVOC. Some novel features in PAH modeling were tested, including seasonality in emissions, the *modal* scheme for particulate-phase tracer representation, the ppLFER scheme for gas−particle partitioning, and re-volatilization from surfaces. The results indicate that using seasonal emission compensates for model biases in the predictions of more volatile species (PHE) whereas the effects of the *modal* and ppLFER schemes are of less significance. Re-volatilization increases the near-ground concentrations in air, which is found most significant for species of mid semivolatility (PYR and FLT). Hereby attribution of model response to individual features (factors) is blurred by the non-linear interactions between two and more factors. The effects of these interactions are found to both reinforce (positive feedback) and suppress (hence, negative feedback) the effects of the individual factors.

For near-surface concentrations, model bias varies by region and/or species, being negative (positive) in the Arctic within typically a factor of $2-13$ (6% to a factor of two) for PHE and BaP (PYR and FLT), positive in the northern mid-latitudes for PHE, PYR, and FLT by up to a factor of three, negative in the Tropics (by a factor of $2-3$) and largely over ocean up to a factor of three orders of magnitude. The model adequately reproduces the seasonal variation of the particulate mass fraction ($\theta$), but underestimates $\theta$ for high-to-medium volatility PAHs. This might be related to a systematic underestimation of OC by the model, which neglects secondary organic aerosols (SOA). The latter may cause significant underestimation of the overall atmospheric aerosol burden and $\theta$ of SOCs, in particular over ocean. Since recently a MESSy submodel, ORACLE, dedicated to the simulations of SOA (Tsimpidi et al., 2014) based on lumping organic species in volatility bins is available. It should be included in future SOC simulations using EMAC.

Moreover, SVOC implicit assumption of instantaneous gas$-$particle equilibrium may cause both over- and underestimates of $\theta$, as inter-phase mass transfer my be kinetically limited to gaseous sources (hence, overestimate of $\theta$) or within the particle bulk (hence, underestimate $\theta$), as the PAHs may become trapped within particles during transport (Friedman et al., 2014; Zelenyuk et al., 2012; Mu et al., 2018). For multidecadal studies, the coupling of a 3D ocean model (coupled with a marine biogeochemistry module) would be needed since the present model application does not allow for horizontal and vertical transports in the deep ocean. For the same reason, contaminant remobilization within deep soil layers should also be introduced. To this end, a multi-layer (3D) soil compartment would be needed to replace the 2D soil compartment used here.

## Code availability

SVOC submodel presented here has been based on the Modular Earth Submodel System (MESSy) version 2.50 and the global atmospheric model ECHAM version 5.3.02. MESSy is continuously further developed and applied by a consortium of institutions. The usage of MESSy and access to the source code is licensed to all affiliates of institutions which are members of the MESSy Consortium. Institutions can be a member of the MESSy Consortium by signing the MESSy Memorandum of Understanding. More information can be found on the MESSy Consortium website (http://www.messy-interface.org). The SVOC submodel will be incorporated into the next released version of the ECHAM/MESSy (EMAC) model (v2.55) and will therefore be made publicly available (with respect to the EMAC license regulations).

## Author contributions

M.O. and G.L. conceived the study and designed the experiments. M.O. developed the SVOC submodel with input from all co-authors. M.O. performed model simulations and data analyses. M.O and G.L discussed the results. M.O. wrote the manuscript with contributions from all co-authors.

## Acknowledgements

This study was supported by the Max Planck Institute for Chemistry. We thank the MESSy
community and MESSy submodel developers for providing technical support. The model
simulation was performed at the Max Planck Computing and Data Facility (MPCDF),
Garching.

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

# Figures and Tables

## Table 1. Summary of MESSy process submodels used in the study

| Submodel | Purpose | Reference |
|---|---|---|
| AEROPT | Aerosol optical properties | Lauer et al. (2007) |
| AIRSEA | Air−sea exchange | Pozzer et al. (2006) |
| CLOUD | ECHAM5 cloud and precipitation scheme as MESSy submodel | Roeckner et al. (2006) and references therein |
| CONVECT | Convection parameterizations | Tost et al. (2006b, 2010) |
| CVTRANS | Convective tracer transport | Tost (2006) |
| DDEP | Dry deposition of gases and aerosols | Kerkweg et al. (2006a) |
| GMXe | Aerosol dynamics and thermodynamics | Pringle et al. (2010) |
| JVAL | Rate of photolysis | based on Landgraf and Crutzen (1998) |
| LNOX | $NO_x$ production from lightning | Tost et al. (2007) |
| MECCA | Tropospheric and stratospheric chemistry | Sander et al. (2011) |
| OFFEMIS | Offline emissions | Kerkweg et al. (2006b) |
| ONEMIS | Online emissions | Kerkweg et al. (2006b) |
| RAD | ECHAM5 radiation scheme as MESSy submodel | Roeckner et al. (2006); Jöckel et al. (2006) |
| SCAV | Scavenging of gases and aerosols | Tost et al. (2006a) |
| SEDI | Aerosol sedimentation | Kerkweg et al. (2006a) |

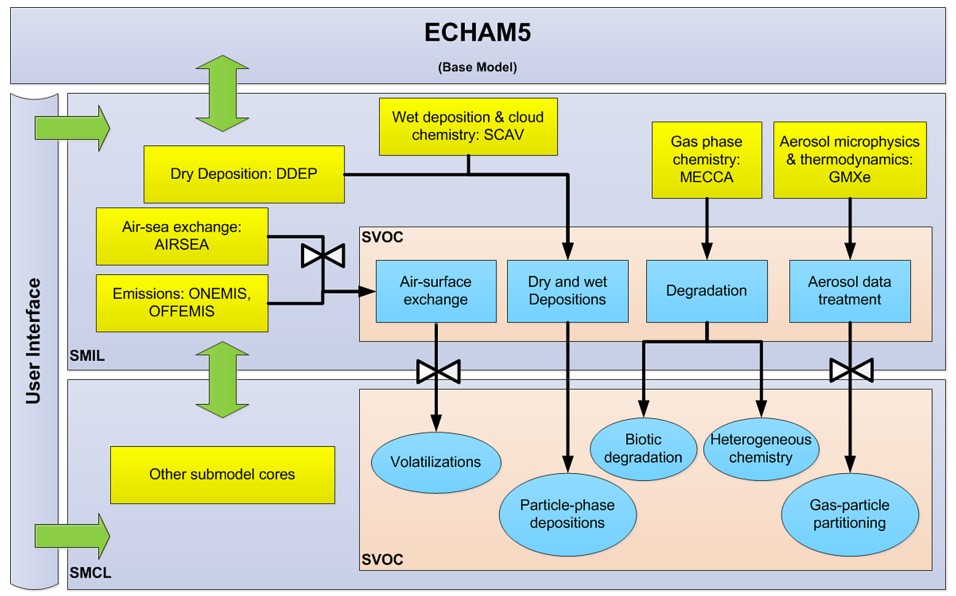

**Figure 1.** Overview of EMAC-SVOC model structure, the cycling processes in SVOC submodel and its interaction with other MESSy submodels. SMIL (submodel interface layer) and SMCL (submodel core layer) are components of MESSy coding standard, see (Jöckel et al., 2006) for further details.

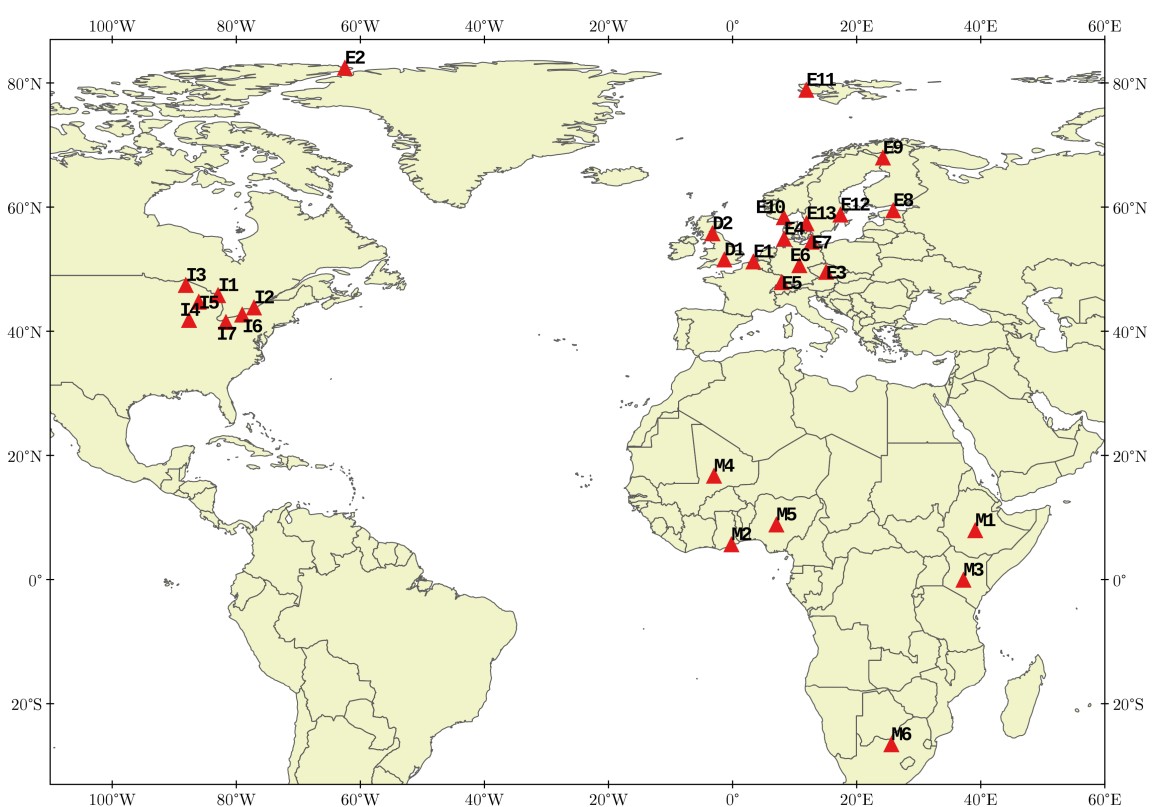

**Figure 2.** Locations of monitoring stations used in the study. The initial letter of each station ID refers to the individual monitoring network (E: EMEP & AMAP; D: DEFRA, I: IADN, M: MONET-Africa)

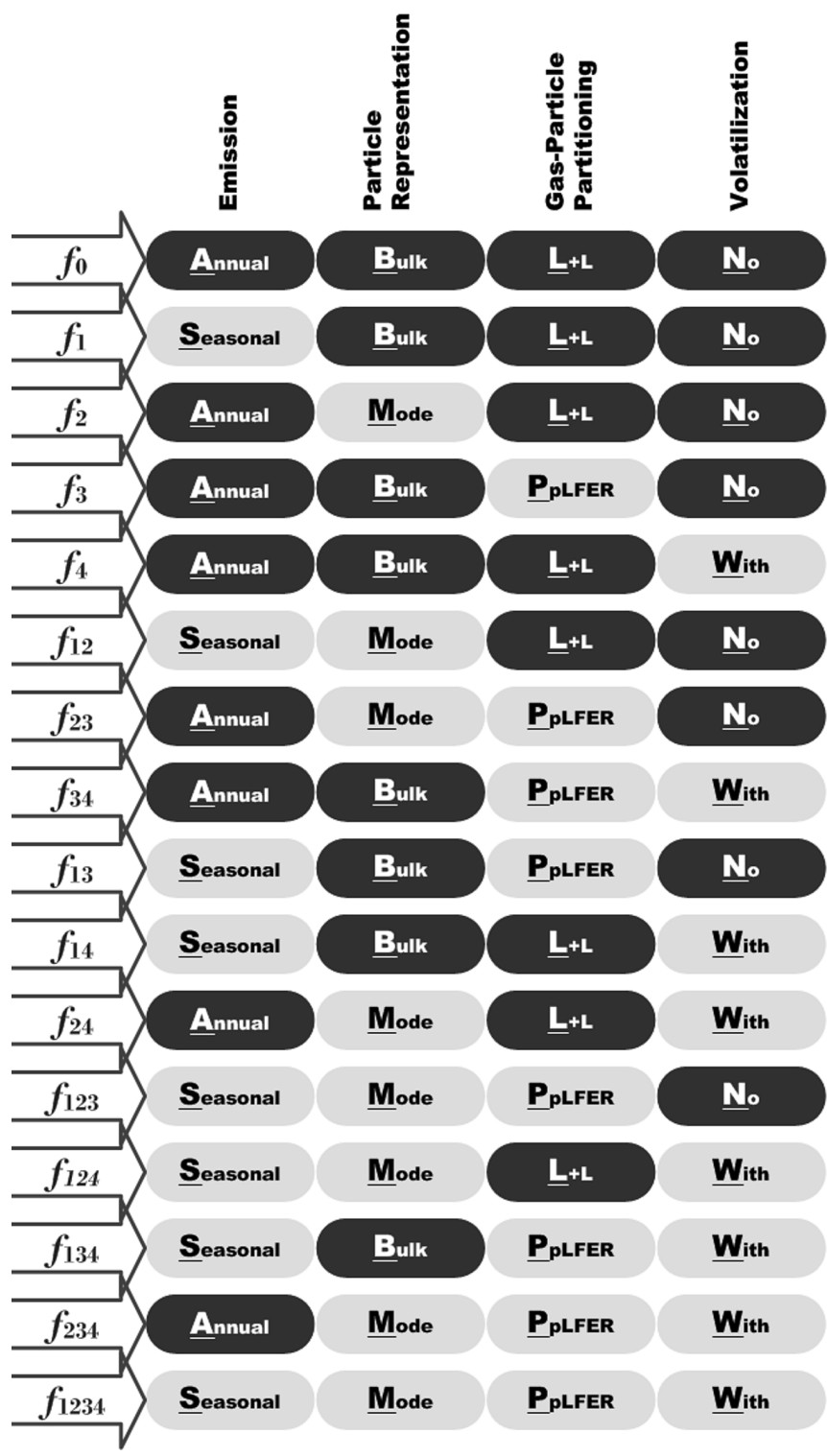

**Figure 3. List of experiments performed for the factor separation analysis to study sensitivity to temporal variation in emission and process parameterizations (particulate-phase representation, gas−particle partitioning scheme, and volatilization). L+L: Lohmann−Lammel, ppLFER: Poly Parameter Linear Free Energy Relationships.**

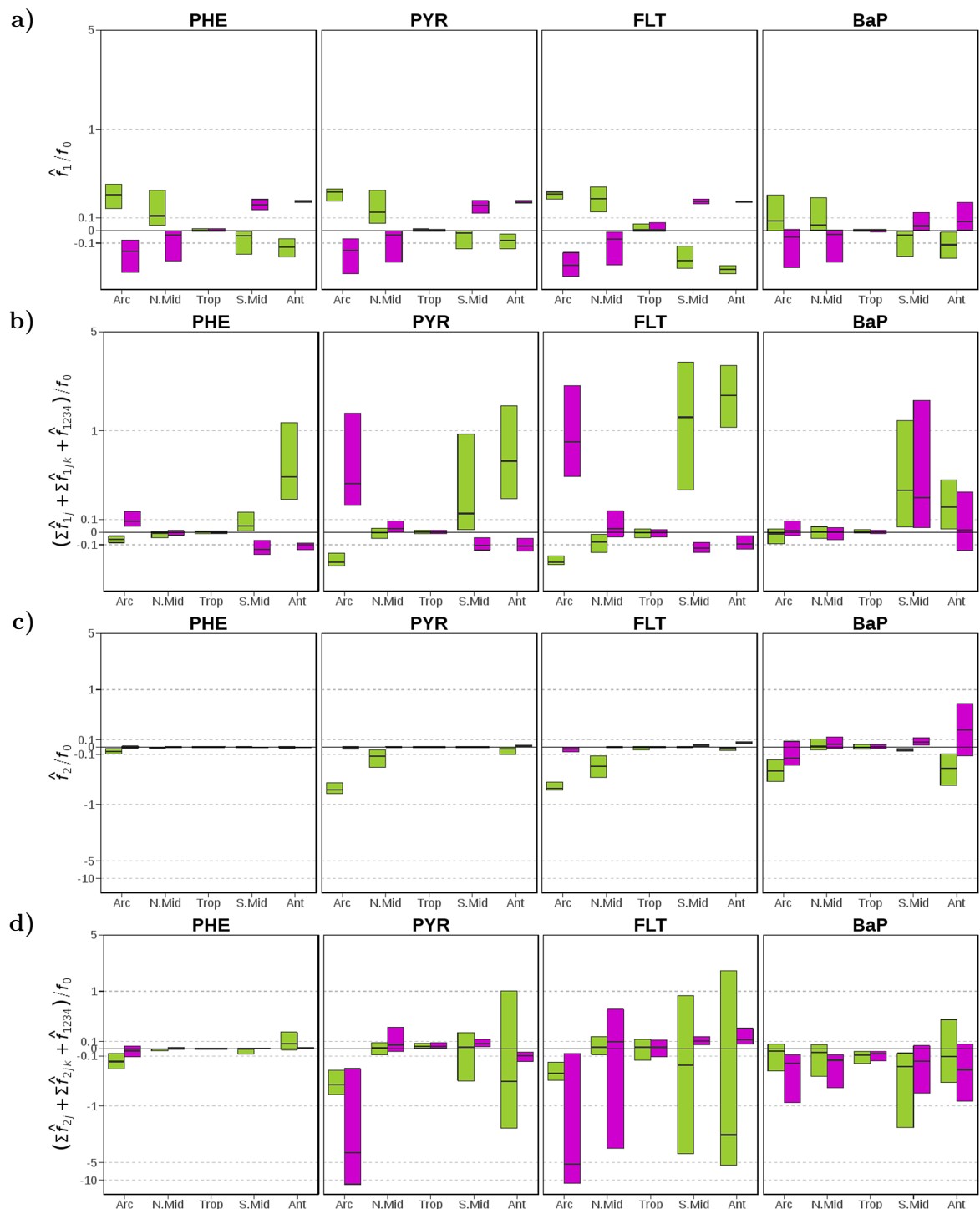

**Figure 4.** Direct and interaction effects on seasonal-mean near-surface PAH concentrations of (a,b) monthly emissions ($i = 1$), (c,d) the *modal* scheme ($i = 2$), (e,f) the ppLFER scheme ($i = 3$), and (g,h) volatilization ($i = 4$). The direct effects (a,c,e,g) are expressed as the difference between two distributions ($\hat{f}_i = f_i - f_0$) whereas the interaction effects (b,d,f,h) are expressed as the sum of two ($\Sigma \hat{f}_{ij}$, $i \neq j$), three ($\Sigma \hat{f}_{ijk}$, $i \neq j \neq k$), and all ($\hat{f}_{1234}$) factor interactions. They are presented as relative to concentrations from the *base* ($f_0$) simulation. The figures display the median, $25^{\text{th}}$ and $75^{\text{th}}$ percentiles of the relative effects over each of five main climatic regions. Note the inverse hyperbolic sine function has been used in scaling the $y$ axes.

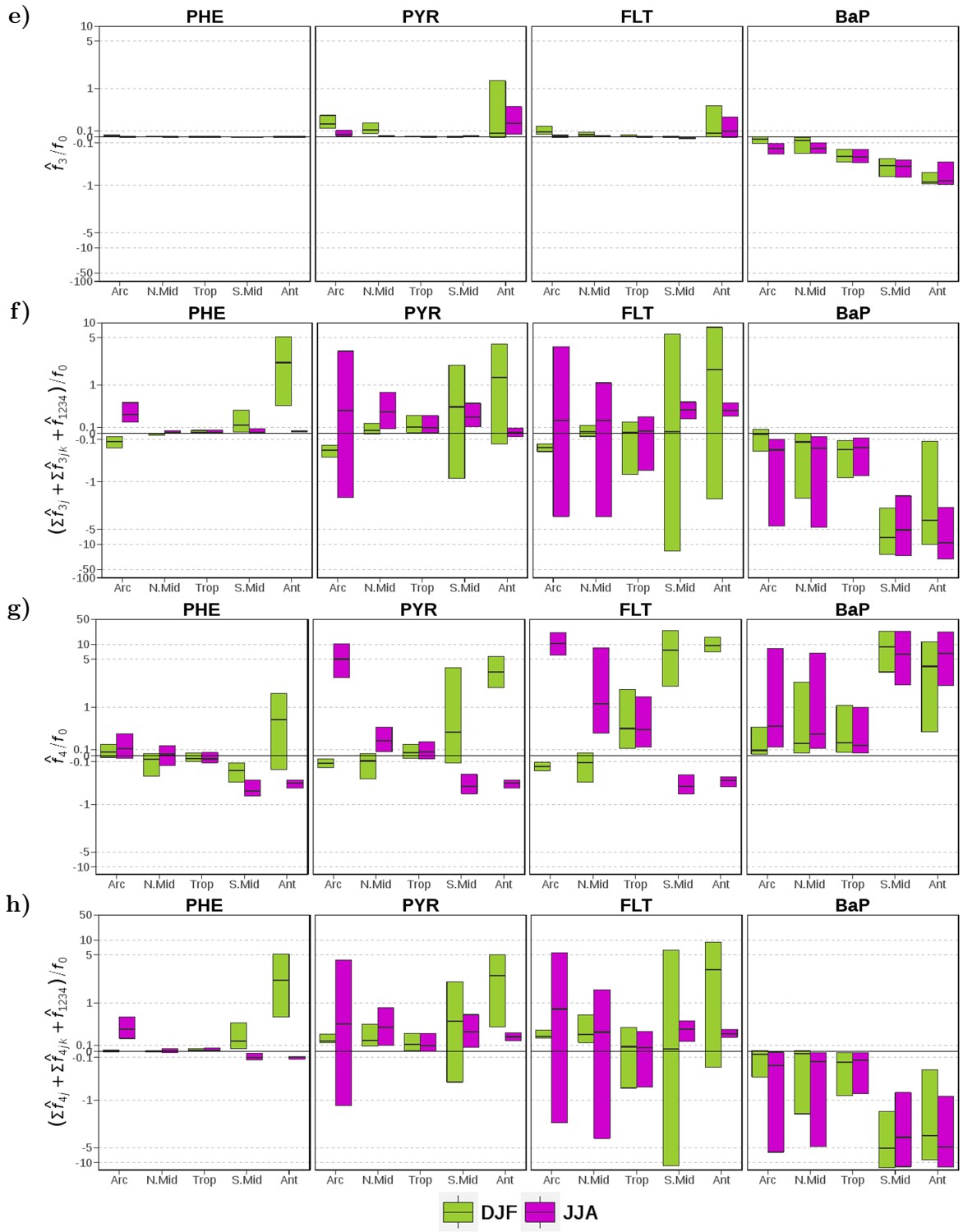

**Figure 4.** continued

**Table 2. Statistics comparison of model simulation and observations of total (gas+particle) concentrations of PAHs from stations in the Arctic, northern mid-latitudes and tropics.** $N$: Number of observed-simulated monthly data pairs; $\overline{x}$: Mean; $Q2_x$: Median; $SD_x$: Standard deviation; $GM_x$: Geometric mean; $x$: Simulated ($M$) or Observed ($O$) data; MB: Mean bias; RMSE: Root mean square error; NMB: Normalized mean bias; NMBF: Normalized mean bias factor; FAC2: Factor of 2; FAC10: Factor of 10; $r$: Correlation coefficient.

| Metrics | Unit | Arctic | | | | NH mid-latitudes | | | | Tropics | | |
|---|---|---|---|---|---|---|---|---|---|---|---|---|
| | | PHE | PYR | FLT | BaP | PHE | PYR | FLT | BaP | PHE | PYR | FLT |
| $N$ | months | 89 | 89 | 89 | 46 | 361 | 328 | 372 | 405 | 34 | 34 | 34 |
| $N_{<\mathrm{LOQ}}$ | months | 0 | 0 | 0 | 30 | 0 | 0 | 0 | 0 | 0 | 0 | 0 |
| $\overline{O}$ | ng m$^{-3}$ | 0.107 | 0.024 | 0.039 | 0.007 | 2.193 | 0.408 | 0.803 | 0.141 | 11.818 | 6.431 | 6.843 |
| $Q2_O$ | ng m$^{-3}$ | 0.034 | 0.014 | 0.012 | 0.002 | 1.301 | 0.194 | 0.360 | 0.037 | 3.608 | 2.106 | 2.181 |
| $SD_O$ | ng m$^{-3}$ | 0.162 | 0.027 | 0.054 | 0.015 | 2.956 | 0.582 | 1.135 | 0.253 | 16.598 | 10.141 | 10.217 |
| $GM_O$ | ng m$^{-3}$ | 0.051 | 0.014 | 0.018 | 0.003 | 0.968 | 0.221 | 0.383 | 0.046 | 3.733 | 1.369 | 1.726 |
| $\overline{M}$ | ng m$^{-3}$ | 0.046 | 0.025 | 0.079 | 5.2E−4 | 2.270 | 1.086 | 1.670 | 0.059 | 4.966 | 2.005 | 3.400 |
| $Q2_M$ | ng m$^{-3}$ | 0.010 | 0.007 | 0.034 | 1.9E−5 | 0.840 | 0.500 | 0.736 | 0.022 | 4.274 | 1.236 | 2.012 |
| $SD_M$ | ng m$^{-3}$ | 0.089 | 0.040 | 0.099 | 6.5E−4 | 2.955 | 1.225 | 2.007 | 0.085 | 3.897 | 2.019 | 3.462 |
| $GM_M$ | ng m$^{-3}$ | 0.012 | 0.008 | 0.041 | 3.4E−5 | 1.144 | 0.635 | 0.913 | 0.028 | 2.816 | 1.038 | 1.788 |
| MB | ng m$^{-3}$ | −0.060 | 0.001 | 0.040 | −0.006 | 0.077 | 0.679 | 0.867 | −0.083 | −6.851 | −4.426 | −3.443 |
| RMSE | ng m$^{-3}$ | 0.118 | 0.038 | 0.099 | 0.016 | 3.564 | 1.404 | 2.383 | 0.279 | 16.005 | 10.631 | 10.392 |
| NMB | - | −0.56 | 0.06 | 1.04 | −0.92 | 0.04 | 1.66 | 1.08 | −0.58 | −0.58 | −0.69 | −0.50 |
| NMBF | - | −1.30 | 0.06 | 1.04 | −12.17 | 0.04 | 1.66 | 1.08 | −1.40 | −1.38 | −2.21 | −1.01 |
| FAC2 | - | 0.20 | 0.28 | 0.30 | 0.17 | 0.40 | 0.39 | 0.30 | 0.31 | 0.26 | 0.24 | 0.29 |
| FAC10 | - | 0.82 | 0.90 | 0.94 | 0.33 | 0.90 | 0.84 | 0.83 | 0.79 | 0.97 | 0.79 | 0.85 |
| $r$ | - | 0.83 | 0.42 | 0.42 | 0.16 | 0.27 | 0.23 | 0.09 | 0.01 | 0.63 | 0.33 | 0.29 |

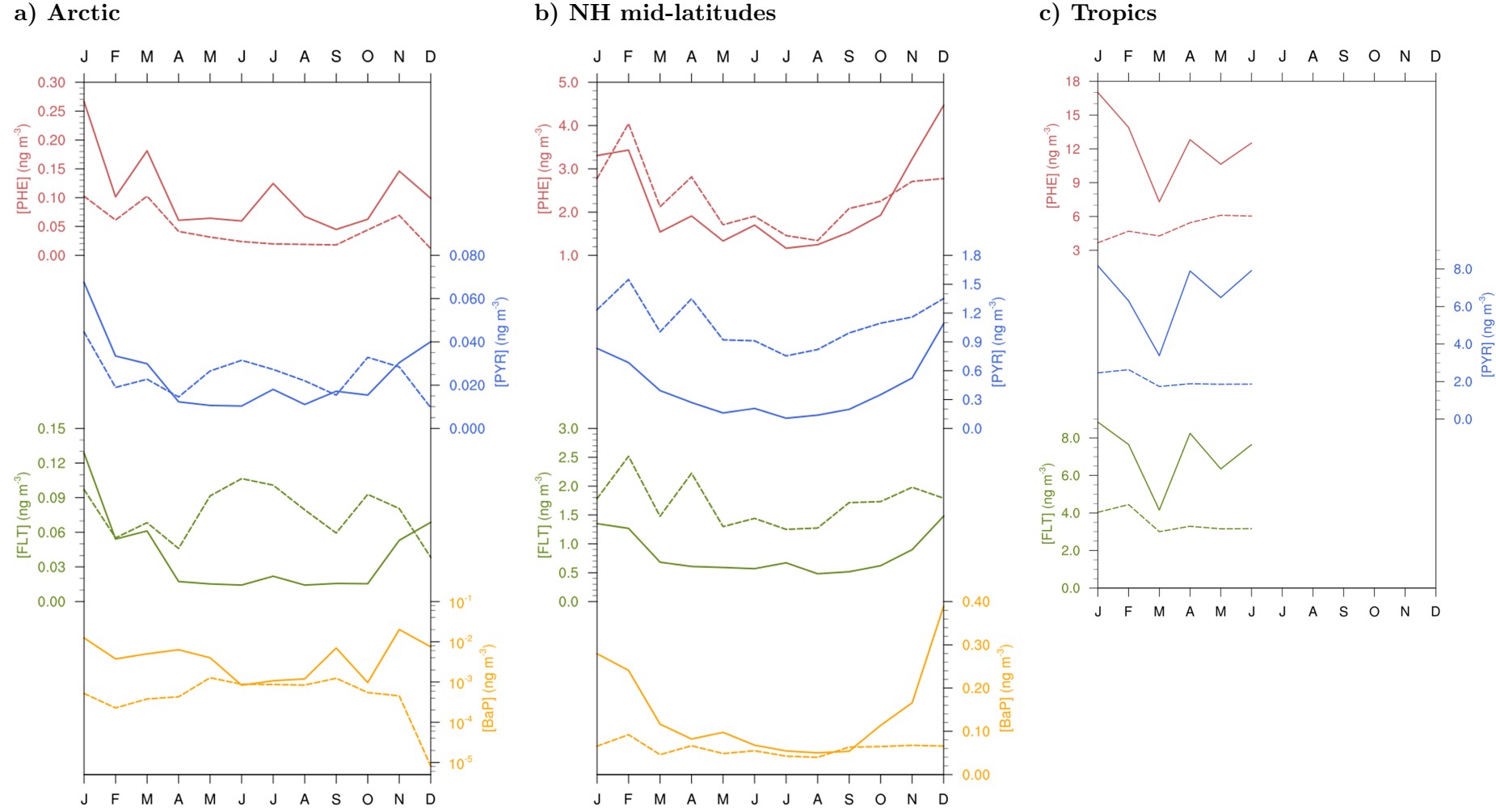

**Figure 5.** Seasonal mean total (gas+particle) concentrations of PAHs (ng m$^{-3}$) from observations (solid lines) and simulations (dashed lines) averaged over all stations in the (a) Arctic, (b) northern mid-latitudes, and (c) tropics. Note that logarithmic scale has been used for BaP concentrations in the Arctic.

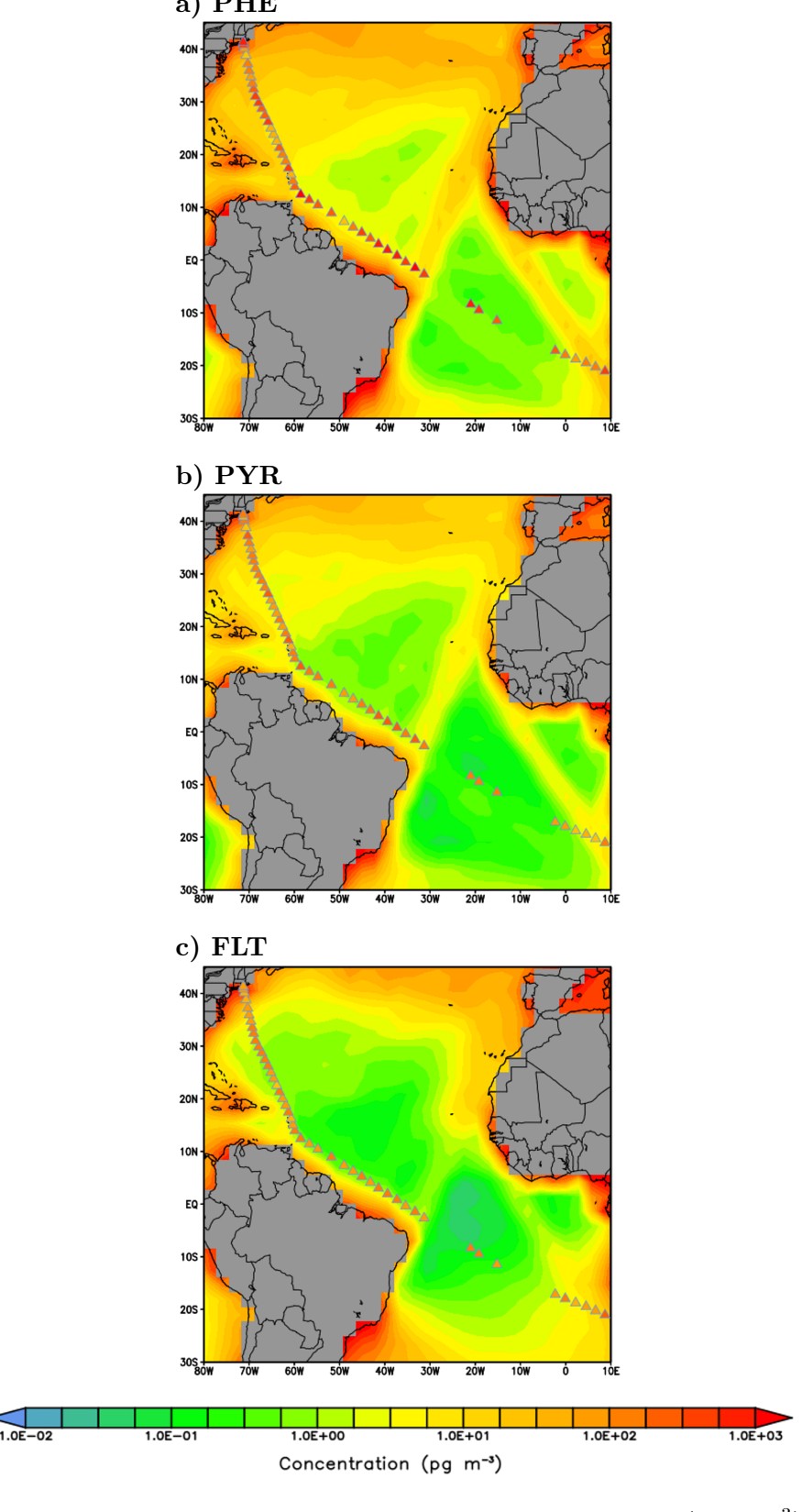

**Figure 6. Simulated concentrations of PHE, PYR, and FLT (pg m$^{-3}$) over the Atlantic ocean overlaid with concentrations from a ship cruise measurement campaign during July 2009 (triangles). Land grid cells are depicted in gray shades.**

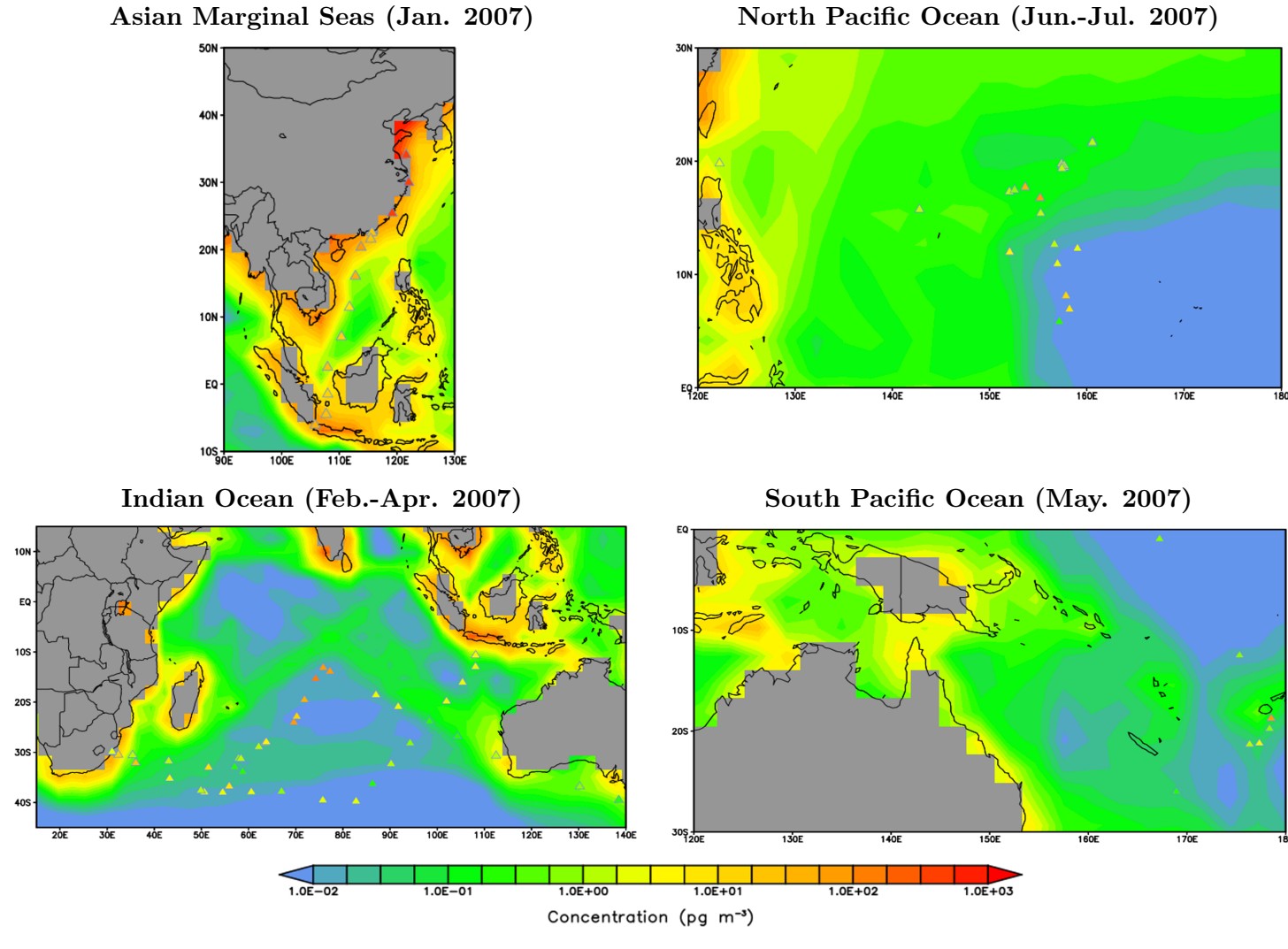

**Figure 7.** Simulated BaP concentrations (pg m$^{-3}$) over the four ocean margins overlaid with concentrations from a ship cruise measurement campaign (triangles). Land grid cells are depicted in gray shades.

**Table 3. Statistics comparison of model simulation and observations of particulate mass fraction ($\theta$) from a subset of surface stations, as listed in Table S11.** $N$: Number of observed-simulated monthly data pairs; $\overline{x}$: Mean; $\mathrm{SD}_x$: Standard deviation; $x$: Simulated ($M$) or Observed ($O$) data; MB: Mean bias; RMSE: Root mean square error; NMB: Normalized mean bias; NMBF: Normalized mean bias factor; FAC2: Factor of 2; FAC10: Factor of 10; $r$: Correlation coefficient.

| Metrics | PHE | PYR | FLT | BaP |
|---|---|---|---|---|
| $N$ | 63 | 63 | 99 | 93 |
| $\overline{O}$ | 0.051 | 0.359 | 0.268 | 0.949 |
| $\mathrm{SD}_O$ | 0.035 | 0.150 | 0.162 | 0.067 |
| $\overline{M}$ | 0.005 | 0.212 | 0.106 | 0.964 |
| $\mathrm{SD}_M$ | 0.005 | 0.138 | 0.086 | 0.027 |
| MB | $-0.046$ | $-0.147$ | $-0.162$ | 0.015 |
| RMSE | 0.057 | 0.214 | 0.225 | 0.074 |
| NMB | $-0.910$ | $-0.410$ | $-0.604$ | 0.016 |
| NMBF | $-10.145$ | $-0.694$ | $-1.523$ | 0.016 |
| FAC2 | 0.00 | 0.56 | 0.30 | 1.00 |
| FAC10 | 0.38 | 1.00 | 0.94 | 1.00 |
| $r$ | 0.42 | 0.42 | 0.33 | 0.03 |

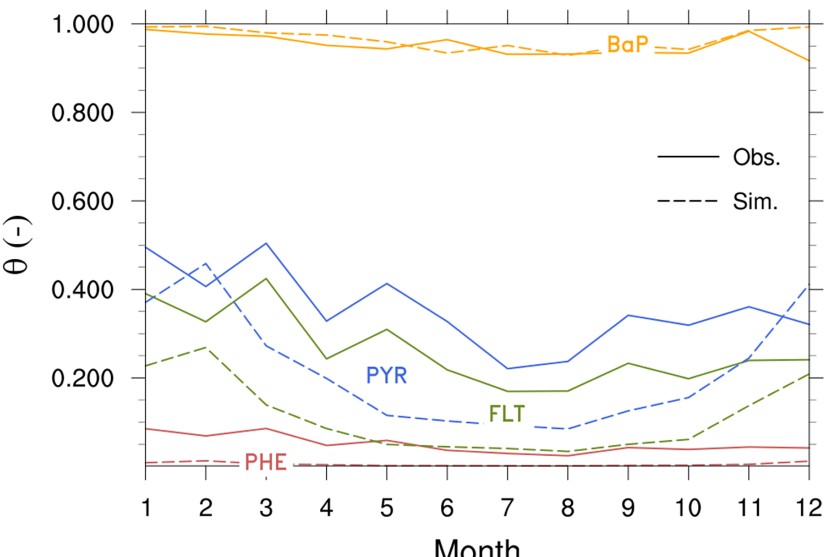

**Figure 8. Seasonal mean particulate mass fraction ($\theta$; unitless) from observations (solid lines) and simulations (dashed lines)**