# Peer review of "Global Simulation of Semivolatile Organic Compounds – Development and Evaluation of the MESSy Submodel SVOC (v1.0)"

_Geoscientific Model Development, 2019_

## Short Comment (SC1) · 14 Mar 2019

Dear authors,

In my role as Executive editor of GMD, I would like to bring to your attention our Editorial version 1.1:

http://www.geosci-model-dev.net/8/3487/2015/gmd-8-3487-2015.html

This highlights some requirements of papers published in GMD, which is also available on the GMD website in the 'Manuscript Types' section:

http://www.geoscientific-model-development.net/submission/manuscript_types.html

[Figure]

In particular the following requirements is not fully met in the Discussions paper:

- "All papers must include a section, at the end of the paper, entitled 'Code availability'. Here, either instructions for obtaining the code, or the reasons why the code is not available should be clearly stated. It is preferred for the code to be uploaded as a supplement or to be made available at a data repository with an associated DOI (digital object identifier) for the exact model version described in the paper. Alternatively, for established models, there may be an existing means of accessing the code through a particular system. In this case, there must exist a means of permanently accessing the precise model version described in the paper. In some cases, authors may prefer to put models on their own website, or to act as a point of contact for obtaining the code. Given the impermanence of websites and email addresses, this is not encouraged, and authors should consider improving the availability with a more permanent arrangement. After the paper is accepted the model archive should be updated to include a link to the GMD paper."

Last summer the GMD executive Editors and the MESSy consortium agreed to a procedure that meets the GMD requirements as well as the MESSy code development standards. The MESSy website (www.messy-interface.org → License) states (among others) the following: "As the exact code described and used in the paper needs to published, it needs to be ensured, that exactly the code published is part of the next official relase (version Y). This requires, that the code is checked in by the developer and approved by the source code administrators before publication, or, to be more precise, before starting the simulations analysed in the publication. In case of doubt, please contact the Consortium Steering Group for advice."

To date, the SVOC code has not been received by the MESSy source code administrators and, consequently, could not be approved for the next official MESSy code release (v2.55).

As the permanent availability of the code published in the paper is not yet guaranteed. The paper can not be finally published until the above requirements are met.

Yours,

Astrid Kerkweg (executive editor of GMD)

---

## Referee Comment (RC1) · Anonymous Referee #1 · 8 Apr 2019

General Comments:

This paper addresses the modeling of atmospheric PAH transport and chemistry/physics, which is certainly within the scope of GMD. It addresses some important processes for PAHs which have been, to the best of my knowledge, previously unresolved in models of its kind. This model built into the MESSy framework is a substantial contribution to the modeling of PAHs in the atmosphere. The methods are clearly outlined, and important assumptions are explicitly tested and discussed, leading to a reproducible work. The authors have given due emphasis to the existing literature, and their own contribution is clearly documented. The overall presentation of the paper

is good, including language, adherence to title and abstract requirements, and formulae. The supporting information is very strong, but the code corresponding to the work appears to be not immediately accessible.

Specific comments:

L194: Soil density is a parameter of the capacity for PAH uptake. Is this density spatially specified? What is the origin of the value used (spatial database, land model or otherwise)?

L201: Similarly to above, is the fraction of organic carbon in soil spatially varying? What is its origin?

L215-220: Some clarifying statements on the application of volatilization from vegetation are warranted. Particularly, it is not clear to me how this CV is applied. Is the CV at 7 days after application used to calculate a timescale for complete revolatilization? i.e. to fit an exponential return to the atmosphere for deposited PAHs. Or is the fraction not volatilized after 7 days assumed to be permanently deposited? Also, is a single CV applied to all plant types?

L255: The ocean is treated with comparatively little detail. Some discussion of how this could impact the strong bias of the model compared to measurements over the oceans would be informative. (Currently it is simply listed as a possible contributor to the bias)

L289-291: With a second-order representation, a higher value of kOH only suggests OH as the dominant loss pathway if concentrations of all three oxidants are equal. The concentration of OH would be expected to be much lower than the concentration of ozone, however.

L314-316: The assumption of the rate doubling every 10 degrees is presented without reference. An explanation of the rationale behind this number should be included here.

L332: Why are gaseous reactions switched off for BaP?

[Figure]

L434-436: This output was selected as the single output for analysis, but other quantities could be analyzed using the same model experiments. Were any others investigated, and if so do they show similar behavior? I.e. are the factors affecting total PAH concentration representative of the factors affecting other outputs?

L626-630: The CoV is compared between observations and model output. But the observations may not be representative of the same time-variations as the model. From the screening flowchart (Figure S3), it seems likely that many stations' "monthly" observations represent less than a full month's integration. This should make their monthly values more sensitive to synoptic-scale variations than their model counterparts, and much more sensitive to local-scale phenomena (e.g. convective precipitation and subsequent wet removal).

L775-776: Does the underestimate follow the same pattern as SOA concentration? The omission of SOA in the model should be mentioned in the methods section.

Technical corrections:

L417: I believe this should read: "Two options for this factor were tested:"

L435: The word "selected" is repeated.

L453-454: I believe that point (2) should be reworded. ". . . physically interpret." would be better than ". . . physically justify." if I understand correctly.

L462: "are higher" should read "being higher" at the start of this line.

L597: "On the contrary," should read "In contrast,"

L618: "occur in the gas. . ." should be "occurring in the gas. . ."

L645: "in a qualitative agreement" -> "in qualitative agreement"

L714: "ocean shipping and do" -> "ocean shipping and does"

L715: "potential origins" -> "potential point of origin"

---

## Referee Comment (RC2) · Anonymous Referee #2 · 14 May 2019

The paper describes modeling of the physical and chemical processes of PAHs in a global chemical transport model. Correctly modeling PAHs is important in addressing their adverse health impacts on both human being and the ecosystems. The topic is certainly within the scope of GMD and is of interest to the modeling community. The paper is well-written. I would recommend minor revision before accepted for publication.

First of all, the title of the paper is "Global simulation of semivolatile organic compounds . . .". However, it appears that it was particularly developed for modeling PAHs so a more specific title should be used to reflect this. If the model is intended to be

applied for other SVOCs, this should be clearly stated in the manuscript. Parameters for all other SVOCs treated by the model should be described and model evaluations should be performed. Regarding the "global" in the title of the paper, the main text does not even include a figure to show modeled global distributions. Land surface concentrations of PAHs in different regions (Asia, North American, etc.) have been published before so a global distribution plot would allow the reviewer to compare these with previous studies.

The authors did an excellent job of investigating the factors that affect the modeled concentrations. However, in my opinion, this is somewhat an overkill because many parameters in the model detailed treatments carry large uncertainties. For example, the ppLFER scheme requires partitioning coefficients for more aerosol components. Uncertainties in these parameters may lead to a different judge on their impact on the predicted concentrations comparing to the base model.

The authors appear to imply that the model runs with the most sophisticated treatment of the processes gave the best results as they only presented these results in the model evaluation section. I have a few comments here: (1) models with more detailed processes might not provide the best results due to compensating errors in the model. The authors should compare the model performance with the simpler treatment of the processes (e.g. using the Lohmann−Lammel scheme vs. with the ppLFER scheme; using annual emissions vs. seasonal varying emissions). (2) the model was configured at 2.8 degrees horizontal resolutions thus cannot resolve local gradients when the monitors are not in the remote areas that can represent the average concentrations represented by the grid cells. Are the monitors used in the analyses selected to filter out the non-remote sites? (3) What's the model performance of BC, total PM and size-resolved PM? What about gaseous pollutants (O3?) The authors didn't mention these in the manuscript. Without these, it is hard to further understand the bias in the model predictions. (4) the authors have included some discussions on comparing with results with GEOS-Chem. As the resolution, emission inventories, model time spans

are all different, this appears to be of less value and can be considered to move to SI. There is a tendency these days to write overly long papers with I am not a big fan of.

---

## Author Comment (AC1) · 18 Jun 2019

The comment was uploaded in the form of a supplement:
https://www.geosci-model-dev-discuss.net/gmd-2019-19/gmd-2019-19-AC1-supplement.pdf

---

## Author Comment (AC2) · 18 Jun 2019

Journal: GMD
Title: Global Simulation of Semivolatile Organic Compounds –
Development and Evaluation of the MESSy Submodel SVOC(v1.0)
Author(s): Mega Octaviani et al.
MS No.: gmd-2019-19
MS Type: Model Description
Iteration: Revision

**Reply to RC1**

RC: This paper addresses the modeling of atmospheric PAH transport and chemistry/physics, which is certainly within the scope of GMD. It addresses some important processes for PAHs which have been, to the best of my knowledge, previously unresolved in models of its kind. This model built into the MESSy framework is a substantial contribution to the modeling of PAHs in the atmosphere. The methods are clearly outlined, and important assumptions are explicitly tested and discussed, leading to a reproducible work. The authors have given due emphasis to the existing literature, and their own contribution is clearly documented. The overall presentation of the paper is good, including language, adherence to title and abstract requirements, and formulae.

AR: We would like to thank the referee for his/her comments.

RC: The supporting information is very strong, but the code corresponding to the work appears to be not immediately accessible.

AR: SVOC will be available in the next official MESSy code release (version 2.55). Unfortunately due to the MESSy license conditions, we are unable to release the SVOC source codes as a publicly available electronic supplement. To access the codes, users are required to comply with the MESSy license and user agreement. More information is available on the MESSy Consortium website (https://www.messy-interface.org).

RC: L194: Soil density is a parameter of the capacity for PAH uptake. Is this density spatially specified? What is the origin of the value used (spatial database, land model or otherwise)?

AR: This study applied the global soil density data from Dunne and Willmott (1996) and the soil organic matter content from Batjes (1996). Both are available at a 0.5-degree grid resolution. These soil parameters vary spatially but not temporally. Soil density was based on sorptivity (Dunne and Willmott, 1996) whereas the organic matter content was based on regional measurements and empirical methods (Batjes, 1996). The two references have been added in the revised manuscript (see Line 387).

Dunne K.A., Willmott C.J. (1996) Global distribution of plant-extractable water capacity of soil, Int. J. Climatology 16, 841-859.

Batjes N.H. (1996) Total carbon and nitrogen in the soils of the world, Europ. J. Soil Sci. 47, 151-163.

RC: L201: Similarly to above, is the fraction of organic carbon in soil spatially varying? What is its origin?

AR: See above

RC: L215-220: Some clarifying statements on the application of volatilization from vegetation are warranted. Particularly, it is not clear to me how this CV is applied. Is the CV at 7 days after application used to calculate a timescale for complete revolatilization? i.e. to fit an exponential return to the atmosphere for deposited PAHs. Or is the fraction not volatilized after 7 days assumed to be permanently deposited? Also, is a single CV applied to all plant types?

AR: The volatilization rate from vegetation surfaces decreases with time. The rate obeys an exponential-time law and is determined by vapor pressure. The volatilization parameterization is based on fitting an empirical equation to observations of cumulated volatilizational losses of numerous pesticides over a period of 7 days after applications (Smit et al., 1998). In the model, this parameterization is applied for all plant types. The non-volatilized fraction is deposited (and subsequently accumulated and/or degraded).

Smit A.A.M.F.R., Leistra M., van den Berg F. (1998) Estimation method for the volatilization of pesticides from plants, Environmental Planning Bureau series 4, DLO Winand Staring Centre, Wageningen, the Netherlands, 101 pp.

RC: L255: The ocean is treated with comparatively little detail. Some discussion of how this could impact the strong bias of the model compared to measurements over the oceans would be informative. (Currently it is simply listed as a possible contributor to the bias)

AR: We thank the referee for the suggestion. We tried to add such information to improve the discussions. Additional texts are in:

- Subsection 2.2.5 Lines 260-267: "Sorption of SOCs in water to suspended particulate matter (colloidal or sinking detritus) is neglected. Therefore, SOC concentration in surface seawater and, hence, volatilization from sea surface is overestimated, in particular for very lipophilic ($logKow > 6$) substances. This bias is negligible for the substances studied here (PAHs) which are less lipophilic or volatilization is limited by vapor pressure (e.g., benzo(a)pyrene). Forces from strong winds, dissolved or particulate organics in seawater are transferred to air via sea spray, which adds to 265 particulate OM in air over the ocean (O'Dowd et al., 2008; Qureshi et al., 2009). This process is neglected in the model."
- Subsection 3.2.3 Lines 705-707: "BaP, mostly stays in the particulate phase, presumably also in seawater, therefore, may be somewhat underestimated due to the neglect of sea-spray driven aerosol suspension."

Nevertheless, we admit that the impact of neglecting ocean dynamics on model bias is complex and requires a further investigation.

O'Dowd C. D., Langmann B., Varghese S., Scannell C., Ceburnis D., Facchini M. C. (2008) A combined organic-inorganic sea-spray source function, Geophysical Research Letters, 35, L01 801, doi:10.1029/2007GL030331.

Qureshi A., MacLeod M., Hungerbühler K. (2009) Modeling aerosol suspension from soils and oceans as sources of micropollutants to air, Chemosphere, 77, 495–500, doi:10.1016/j.chemosphere.2009.07.051.

RC: L289-291: With a second-order representation, a higher value of kOH only suggests OH as the dominant loss pathway if concentrations of all three oxidants are equal. The concentration of OH would be expected to be much lower than the concentration of ozone, however.

AR: Experimental studies of reaction rates using typical concentrations of atmospheric oxidants indicate that the gas phase reaction with OH remains the dominant loss process for most PAHs (Keyte et al., 2013 and references therein). The average global OH concentration has been estimated to be $1.16 \times 10^6$ molecules cm$^{-3}$ (Spivakovsky et al., 2000). A reasonable ozone concentration range during day-time in the lower troposphere is 20−50 ppbv (at 298 K and 1 atm, $5 \times 10^{11}$−$1 \times 10^{12}$ molecules cm$^{-3}$). The experimental studies have reported small loss of 2-4 ring PAHs during exposure with high ozone concentrations ($\approx 4 \times 10^{13}$ molecules cm$^{-3}$). $k_{O3}$ was reported to be approximately eight orders of magnitude lower than $k_{OH}$.

Keyte I.J, Harrison R.M., Lammel G. (2013) Chemical reactivity and long-range transport potential of polycyclic aromatic hydrocarbons–A review, Chem. Soc. Rev. 42, 9333–9391.

Spivakovsky C.M., Logan J.A., Montzka S.A., Balkanski Y.J., Foreman-Fowler M., Jones D.B.A, et al. (2000) Three-dimensional climatological distribution of tropospheric OH: Update and evaluation. J. Geophys. Res. 105, 8931-8980.

RC: L314-316: The assumption of the rate doubling every 10 degrees is presented without reference. An explanation of the rationale behind this number should be included here.

AR: Following the referee's suggestion, the following sentence was added (see Lines 328-331): "The 10 K temperature warming is assumed to double the rate of degradation, following recommendations in chemicals risk assessment (European Commission, 2000) and consistent with findings, such as a two-time increase in the growth of hydrocarbon-degrading microbes in soils (Thibault and Elliott, 1979)."

European Commission: Guidance document on persistence in soil (2000), Technical Report 9188/VI/97 in relation to Council Directive No. 97/57/EC, EC Directorate General for Agriculture.

Thibault G.T., Elliott N.W. (1979) Accelerating the biological cleanup of hazardous materials spills. In Proc. Oil and Haz. Mater. Spills: Prevention-Control-Cleanup-Recovery-Disposal.

RC: L332: Why are gaseous reactions switched off for BaP?

AR: There is little significance of gaseous oxidation of BaP, as this substance mostly stays in the particulate phase. In addition, the reaction rate of BaP homogeneous oxidation remains poorly studied. This is mainly due to the experimental difficulty of studying the substance in the gas phase. Henceforth, the gaseous oxidation was switched off.

RC: L434-436: This output was selected as the single output for analysis, but other quantities could be analyzed using the same model experiments. Were any others investigated, and if so do they show similar behavior? I.e. are the factors affecting total PAH concentration representative of the factors affecting other outputs?

AR: The present study tries to quantify the response of model predictions to variation in emission interval, particulate-phase representation, the choice of gas-particle partitioning scheme, and volatilization. It is obvious that the changes in one or a few of these factors will cause a change in all model outputs. However, since emissions and volatilization are applied only at the surface, their direct effects may have little relevance to some of the outputs (e.g., atmospheric burden). We decided to perform the sensitivity analysis for near-surface concentrations, to allow us to evaluate the range of model bias in comparison to observations. However, in Section 3.1, we also presented shortly the direct effects of some of the factors to total deposition, partitioning coefficient, and lifetime, in order to support our discussion (see Lines 512-513 and Lines 555-559).

RC: L626–630: The CoV is compared between observations and model output. But the observations may not be representative of the same time-variations as the model. From the screening flowchart (Figure S3), it seems likely that many stations' "monthly" observations represent less than a full month's integration. This should make their monthly values more sensitive to synoptic-scale variations than their model counterparts, and much more sensitive to local-scale phenomena (e.g. convective precipitation and subsequent wet removal).

AR: We thank the referee for sharing his/her view. It is true that the monthly observation data at a few stations might be representative only for several days. Hence, they could be strongly linked to specific synoptic- and local-scale phenomena. Such influences would be less apparent in the model since the monthly mean values are used. However, the observations in Table 2 are represented as regional values, derived from pooling data from all stations in each region. This makes the impacts of synoptic- and local-scale variations associated with a specific station to the pooled data become less dominant, as the spatial extent increases and the time range varies across stations. Note that the ECMWF reanalysis dataset was applied to nudge the model meteorology, allowing the model to reflect the observed large-scale dynamics. Due to the reasons above, we may conclude that the CoV comparison is compatible and the differences between the model and observations are likely more model intrinsic than being induced by the time integration.

RC: L775–776: Does the underestimate follow the same pattern as SOA concentration? The omission of SOA in the model should be mentioned in the methods section.

AR: We thank the referee for the suggestion. The following sentence has been added: "It is noteworthy that the formation of secondary organic aerosols (SOA) from atmospheric oxidation and condensation of volatile organic compounds (VOCs) were not treated in the simulations. In the model, particulate organic matter is emitted and transported as a bulk aerosol species (OM)." (see Lines 371-374). PAHs may become trapped in OM, preventing them from evaporating to the gas phase and shielding them from chemical degradation. The omission of SOA as a source of OM is one contributing factor to the negative bias in predicted concentrations and particulate mass fraction. However, the underestimation may not follow the same distribution as SOA concentration. The multiphase reactivity of PAHs depends strongly on the phase state of organic coatings which is controlled by environmental

parameters such as temperature and relative humidity. Global simulations from Shiraiwa et al. (2017) indicate that SOA are mostly liquid in tropical, semi-solid in the mid-latitudes, and glassy solid in the middle and upper troposphere. The effectiveness of shielding by SOA is thus higher at cold and temperate regions, as well as in free and upper troposphere. These regions are not necessarily correlated with SOA-rich environments. The influence of this phase state on global predictions of BaP was discussed in a follow-up study by Mu et al. (2018).

Shiraiwa M., Li Y., Tsimpidi A.P., et al. (2017) Global distribution of particle phase state in atmospheric secondary organic aerosols. Nat. Commun. 8, 15002.

Mu Q., Shiraiwa M., Octaviani M., Ma N., Ding A., Su H., Lammel G., Pöschl U., and Cheng Y. (2018) Temperature effect on phase state and reactivity controls atmospheric multiphase chemistry and transport of PAHs, Sci. Adv. 4, doi:10.1126/sciadv.aap7314.

RC: L417: I believe this should read: "Two options for this factor were tested:"

AR: The sentence has been corrected (see Line 433).

RC: L435: The word "selected" is repeated.

AR: We removed "as the selected output" in the revised manuscript (see Line 451).

RC: L453-454: I believe that point (2) should be reworded. "...physically interpret." would be better than "...physically justify." if I understand correctly.

AR: We followed the referee's suggestion (see Line 469).

RC: L462: "are higher" should read "being higher" at the start of this line.

AR: The sentence has been corrected (see Line 477).

RC: L597: "On the contrary," should read "In contrast,"

AR: The sentence has been corrected (see Line 613).

RC: L618: "occur in the gas..." should be "occurring in the gas..."

AR: The sentence has been corrected (see Line 633).

RC: L645: "in a qualitative agreement" -> "in qualitative agreement"

AR: The sentence has been corrected (see Line 660).

RC: L714: "ocean shipping and do" -> "ocean shipping and does"

AR: The sentence has been corrected (see Line 698).

RC: L715: "potential origins" -> "potential point of origin"

AR: The sentence has been corrected (see Line 699).

**Reply to RC2**

RC: The paper describes modeling of the physical and chemical processes of PAHs in a global chemical transport model. Correctly modeling PAHs is important in addressing their adverse health impacts on both human being and the ecosystems. The topic is certainly within the scope of GMD and is of interest to the modeling community. The paper is well-written. I would recommend minor revision before accepted for publication.

AR: We thank the referee for his/her time and comments. Here we addressed all the comments.

RC: First of all, the title of the paper is "Global simulation of semi volatile organic compounds...". However, it appears that it was particularly developed for modeling PAHs so a more specific title should be used to reflect this. If the model is intended to be applied for other SVOCs, this should be clearly stated in the manuscript. Parameters for all other SVOCs treated by the model should be described and model evaluations should be performed.

AR: SVOC submodel was used here to simulate PAH compounds, as their physicochemical properties are well described and global emission inventory is available. The submodel was developed and is intended to be applied for the study of all potentially re-volatilizing and gas-particle partitioning (hence, semivolatile) compounds (e.g., PCDDs/Fs, PBDEs, novel brominated and phosphoric acid ester flame retardants, Diels-Alder organochlorine compounds) once their properties and emissions become apparent. In the revised manuscript, we included this statement in the introduction (see Lines 66-67). Further modifications might be deemed necessary, such as integrating new and improved parameterizations into SVOC.

RC: Regarding the "global" in the title of the paper, the main text does not even include a figure to show modeled global distributions. Land surface concentrations of PAHs in different regions (Asia, North American, etc.) have been published before so a global distribution plot would allow the reviewer to compare these with previous studies.

AR: The global distributions of PAH concentrations from the *base* experiment ($f_0$: annual emission + the *bulk* scheme + the Lohmann-Lammel scheme + no re-volatilization) are presented Fig. S4. In the revised manuscript, those from the *target* experiment ($f_{1234}$: i.e., seasonal emissions + the *modal* scheme + ppLFER scheme + with re-volatilization) were added (see Fig. S5).

RC: The authors did an excellent job of investigating the factors that affect the modeled concentrations. However, in my opinion, this is somewhat overkill because many parameters in the model detailed treatments carry large uncertainties. For example, the ppLFER scheme requires partitioning coefficients for more aerosol components. Uncertainties in these parameters may lead to a different judge on their impact on the predicted concentrations comparing to the base model.

AR: This study investigated the response of simulated concentrations to changes in the selected model factors. Studying uncertainty in the simulated concentrations resulting from the uncertainty in model input parameters could be an important factor, but is not an easy task and beyond the scope of this study. One previous work, i.e. Thackray et al. (2016) could be suitably followed for that purpose. The ppLFER model does not need more input data than available in the GMXe, hence, the off-line evaluation (Shahpoury et al 2016) of the parameterization suffices. As described in Section 2.2.2, each partition coefficient requires information on system parameters (Table S2) which are aerosol-system specific and solute descriptors (Table S6) which are substance specific. The partitioning coefficient is then parameterized as a function of temperature (for all aerosol systems) and humidity (for salts only). We argue any related uncertainty associated with the partitioning could lead to a different judge on the ppLFER impact on the predicted concentrations than what has been described in our study.

The accuracy of temporal variation of emissions can also be a source of uncertainty but we assume that they would have a limited effect on our coarse resolution and time averages. For future study, we would suggest more to test the sensitivity to neglect the SOA Formation.

Thackray C. P., Friedman C. L, Zhang Y, Selin N. E. (2015) Quantitative assessment of parametric uncertainty in Northern Hemisphere PAH concentrations. Environ. Sci. Technol. 49 (15), 9185-9193.

Shahpoury P., Lammel G., Albinet A., Sofuoğlu A., Domanoğlu Y., Sofuoğlu C.S., Wagner Z., Ždimal V. (2016) Evaluation of a conceptual model for gas-particle partitioning of polycyclic aromatic hydrocarbons using poly-parameter linear free energy relationships. Environ. Sci. Technol. 50, 12312-12319.

RC: The authors appear to imply that the model runs with the most sophisticated treatment of the processes gave the best results as they only presented these results in the model evaluation section. I have a few comments here: (1) models with more detailed processes might not provide the best results due to compensating errors in the model. The authors should compare the model performance with the simpler treatment of the processes (e.g. using the Lohmann–Lammel scheme vs. with the ppLFER scheme; using annual emissions vs. seasonal varying emissions).

AR: The influence of sophistication of gas-particle partitioning model, temporal resolution of emissions, and other features/parameterizations, has been tested and discussed in Section 3.1. In Section 3.2 (Model Evaluation), we only evaluated the model results from the *target* experiment to assess the current state of model predictive capability using the most recent knowledge in PAH modeling. On the other hand, Section SVII describes the ranges of model bias obtained from all sensitivity experiments, which is useful to compare model performance from each combination against the *base* configuration simulation (referring to the referee's suggestion). The results vary depending on species, region, and season and it appears difficult to state explicitly if the sophisticated treatment of the processes performs best for all cases. For example: (1) using seasonal emissions brings predictions closer to observations for PHE concentrations, particularly in the northern mid-latitudes; (2) the ppLFER scheme yields smaller Arctic concentrations than the Lohmann-Lammel scheme, leading to higher negative

bias in this region; (3) volatilization would reduce negative bias in BaP concentrations during summer, but may lead to a positive bias in combination with the Lohmann-Lammel scheme.

RC: (2) the model was configured at 2.8 degrees horizontal resolutions thus cannot resolve local gradients when the monitors are not in the remote areas that can represent the average concentrations represented by the grid cells. Are the monitors used in the analyses selected to filter out the non-remote sites?

AR: The referee is correct, we ignored stations that have close proximity to urban and industrial sources, such as all urban sites. Only data from rural background or remote stations were screened and quality checked (following the description in Section SIII).

RC: (3) What's the model performance of BC, total PM and size-resolved PM? What about gaseous pollutants (O3?) The authors didn't mention these in the manuscript. Without these, it is hard to further understand the bias in the model predictions.

AR: Model performance for global distributions of aerosols and gaseous oxidants has been confirmed in previous studies using the EMAC model. The GMXe aerosol microphysics and gas-aerosol partitioning submodel implemented within EMAC has been shown to improve model predictions of various aerosol species, including BC, OM, dust, sea salt, sulfate, ammonium, and nitrate aerosol (Pringle et al., 2010). The simulated values show generally a good agreement with observations for the bulk aerosol species, particularly for BC and OM where at least 90% of modeled values are within a factor of two of the observations. For tropospheric ozone, it has been shown that EMAC reproduces the annual cycle and spatial pattern of observed tropospheric column ozone (Righi et al., 2015). However, the model tends to overestimate the magnitude, in particular over the NH mid-latitudes.

Pringle K. J., Tost H., Message S., Steil B., Giannadaki D., Nenes A., Fountoukis C., Stier P., Vignati E., and Lelieveld, J. (2010) Description and evaluation of GMXe: a new aerosol submodel for global simulations (v1), Geosci. Model Dev. 3, 391–412

Righi M., Eyring V., Gottschaldt K.-D., Klinger C., Frank F., Jöckel P., and Cionni, I (2015) Quantitative evaluation of ozone and selected climate parameters in a set of EMAC simulations, Geosci. Model Dev. 8, 733-768.

RC: (4) the authors have included some discussions on comparing with results with GEOS-Chem. As the resolution, emission inventories, model time spans are all different, this appears to be of less value and can be considered to move to SI. There is a tendency these days to write overly long papers with I am not a big fan of.

AR: We followed the referee's suggestion. In the revised manuscript, the intermodel comparison in Section 3.2.1 has been moved to Supporting Information (see Section SIX and Lines 669-670).

**Reply to SC1**

In my role as Executive editor of GMD, I would like to bring to your attention our Editorial version 1.1:

http://www.geosci-model-dev.net/8/3487/2015/gmd-8-3487-2015.html

This highlights some requirements of papers published in GMD, which is also available on the GMD website in the 'Manuscript Types' section:
http://www.geoscientific-model-development.net/submission/manuscript_types.html

In particular the following requirements is not fully met in the Discussions paper:
"All papers must include a section, at the end of the paper, entitled 'Code availability'. Here, either instructions for obtaining the code, or the reasons why the code is not available should be clearly stated. It is preferred for the code to be uploaded as a supplement or to be made available at a data repository with an associated DOI (digital object identifier) for the exact model version described in the paper. Alternatively, for established models, there may be an existing means of accessing the code through a particular system. In this case, there must exist a means of permanently accessing the precise model version described in the paper. In some cases, authors may prefer to put models on their own website, or to act as a point of contact for obtaining the code. Given the impermanence of websites and email addresses, this is not encouraged, and authors should consider improving the availability with a more permanent arrangement. After the paper is accepted the model archive should be updated to include a link to the GMD paper."

Last summer the GMD executive Editors and the MESSy consortium agreed to a procedure that meets the GMD requirements as well as the MESSy code development standards. The MESSy website (www.messy-interface.org → License) states (among others) the following: "As the exact code described and used in the paper needs to published, it needs to be ensured, that exactly the code published is part of the next official release (version Y). This requires, that the code is checked in by the developer and approved by the source code administrators before publication, or, to be more precise, before starting the simulations analysed in the publication. In case of doubt, please contact the Consortium Steering Group for advice."

To date, the SVOC code has not been received by the MESSy source code administrators and, consequently, could not be approved for the next official MESSy code release (v2.55). As the permanent availability of the code published in the paper is not yet guaranteed. The paper can not be finally published until the above requirements are met.

AR: The source codes have been received and approved by the Consortium Steering Group to be part of the next official MESSy release of version 2.55. The corresponding author acts as a reference to provide input data and namelist files applied in the study.